# Strategic Slum Upgrading and Redevelopment Action Plan for Jammu City

**Majid Farooq** [1] **, Fayma Mushtaq** [1] **, Gowhar Meraj** [1] **, Suraj Kumar Singh** [2] **, Shruti Kanga** [3] **, Ankita Gupta** [4] **, Pankaj Kumar** [5] **, Deepak Singh** [6] **and Ram Avtar** [4,7,*]

1   Department of Ecology, Environment & Remote Sensing, Government of Jammu & Kashmir, Srinagar 190018, India
2   Centre for Sustainable Development, Suresh Gyan Vihar University, Jaipur 302017, India
3   Centre for Climate Change and Water Research, Suresh Gyan Vihar University, Jaipur 302017, India
4   Graduate School of Environmental Science, Hokkaido University, Sapporo 060-0810, Japan
5   Institute for Global Environmental Strategies, Hayama 240-0115, Japan
6   Research Institute for Humanity and Nature (RIHN), Motoyama 457-4, Kamigamo, Kitaku, Kyoto 603-8047, Japan
7   Faculty of Environmental Earth Science, Hokkaido University, Sapporo 060-0810, Japan
*   Correspondence: ram@ees.hokudai.ac.jp

**Abstract:** Rapid urbanization has led to the emergence of slums in many developing and industrialized nations. It degrades the quality of life and burdens the urban amenities resulting in uneven distribution of slums. The majority of people in the developing world live in squatter settlements and these random gatherings disrupt the economic and social developmental plans of the concerned country. No suitable planning framework has been created for replicability on a considerable scale, despite the fact that slum upgrading is acquiring worldwide importance as a political issue. In recent years Jammu City has witnessed high population growth rates resulting in an uneven provision of urban amenities and a surge in slum areas. This paper focuses on a method-based approach using Management Information System (MIS) and Geographic Information System (GIS) for upgrading slums and recommends a planning outline using the approach formulated by the Government of India under the scheme named "Rajiv Awas Yojna" (RAY). The aim of this study is to assess the status of slums, propose redevelopment plans, and highlight the roles of different planning agencies to accomplish the redevelopment goals. The study concludes by postulating several recommendations for upgrading slums and formulating a framework that can be used in other similar areas for development.

**Keywords:** GIS; informal settlement; redevelopment; upgrading; matrix; MIS

## 1. Introduction

In 1950, there were 752 million people living in metropolitan areas; today, that number has increased to 4.2 billion [1]. According to the United Nations' "World's Urbanization Prospect, 2018," the global urban population is expected to increase from its current level of 55.7 percent to 68 percent by 2050, implying the addition of 2.5 billion people to the urban population in that time. In addition, over 90% of this growth is expected to occur in Asia and Africa [2]. Slums are common in low- and middle-income nations because of fast urbanization, population growth, rapid rural-to-urban migration, low-income planning, economic stagnation, high unemployment rates, natural disasters, armed conflict, and social unrest, which leaves residents without enough access to essential services and infrastructure and leaving many in dire financial straits [3–9]. Urban populations continue to rise, but government institutions rarely keep up with the need for serviced land equipped with the necessities [10,11]. Because of the skyrocketing cost of land in urban regions, low-income families have no choice but to make their homes in undeveloped and underdeveloped

areas (slums) with inadequate infrastructure [11]. South Asian slums have also become an important part of urbanization, a fact highlighted in "The New Urban Agenda" (United Nations Human Settlements Programme [12].

Slums are densely populated neighborhoods that were not considered during the city's initial design phase [13–15]. According to the Indian Census, slums are neighborhoods where people live in conditions that pose a threat to their health and safety, such as dilapidation, overcrowding, poor building design, narrow streets, a lack of ventilation and light, and inadequate sanitation [16]. It has divided the slums into three groups: those that had been notified, those that had been recognised, and those that had been identified. Areas of a city that have been designated as slums in accordance with the Slum Act have been "notified" to the city. Slums that have been "recognised" by the government of a state or union territory but have not been formally designated as such under any law are nevertheless considered "slums" (UT). The Ministry of Housing and Urban Poverty Alleviation GoI defines slums as densely populated places with a minimum of 300 people and roughly 60–70 households living in overcrowded, substandard housing conditions.

Approximately 30 percent of the urban population in developing countries currently resides in slums, which includes about 1 billion people today [3]. If current trends continue, the number of people living in slums will reach 2 billion in 2030 and 3 billion in 2050 [17]. These are locations without adequate sanitation, clean water, land tenure security, housing, or living space [13]. Health, education, child mortality, social and political marginalization, and many other factors are all influenced by the presence of slums [18]. Between 2014 and 2050, the world's urban population is predicted to rise by a total of 37%, with 37% of that growth occurring in India, China, and Nigeria [19]. Records from the 2011 Census show that 22.4% of India's total population lives in slums [20]. Rough estimates show that 17.4 percent of urban Indian homes, or over 200 million people, live in slums [21]. This equates to about 13.7 million households. The central government in India has sponsored numerous initiatives to improve the living conditions of India's urban poor, with the goals of eliminating, legalizing, and redeveloping informal settlements [22]. Rajiv Awas Yojana (RAY), Jawaharlal Nehru National Urban Renewal Mission (JNNURM), Pradhan Mantri Awas Yojana, etc. are just some of the many programmes implemented by the Government of India to remove the slums [23]. In order to facilitate in situ renovation of houses and overall settlement upgrading, these plans have recently centered on mass evictions, relocation of slum and pavement occupants, and other similar initiatives [24]. However, the rate at which unplanned communities are sprouting up is matching that at which urban slums are being cleared. The increasing number of people living in slums poses a threat to the government officials who are tasked with maintaining order there. Rapid changes in the form, size, and conditions of informal settlements leave them especially susceptible to catastrophes such as fire and flood [25]. The land use and ecological sustainability suffers as a result of urbanization. Therefore, cutting-edge methods must be implemented promptly if the idea of sustainable growth is to progress [26]. For achieving the United Nations' Sustainable Development Goals [27], it is a very challenging task in developing nations to provide access to clean water and sanitation. Nearly 30 years of debate on how to best handle these settlements has apparently culminated in widespread consensus among a range of parties, that improving the settlements in situ is the best course of action [28]. Slum upgrading is crucial to achieving the sustainable development aim and bettering the current situation. Slum infrastructure and the quality of life for slum residents will undeniably benefit from this type of modernization. The administration must act swiftly to save the worst slums in this scenario [29].

Prior studies on slum research focused on (a) socio-economic and policy [30–32] and (b) geo-physical characteristics using approaches such as remote sensing [33–35]. Moreover, recent research into slum characteristics has also focused on automatic feature extractions using cellular automata (CA) and agent-based models (ABMs) [36–39]. Water, sanitation, adequate living space, adequate structure, kitchen and cooking fuel, bathrooms, and electricity are just some of the seven criteria that were used to create a slum severity index

in a recent work by [40]. Slum Severity Index (SSI) based on shelter deprivation in Mexico City was also developed, with the exception of [41], which used a more robust approach. Their scale, which is based on the UNHABITAT definition of a slum and on insights from the work of [40], defines a unidimensional measure of SSI that may be adequate for the purposes of their study but is limited in its conceptualization of housing insecurity because it does not consider security of tenure and/or the deficiencies in the built environment of slums. However, other studies have narrowed their focus to one aspect of housing insecurity in order to measure it, whether that be tenure security, the likelihood of eviction, or property rights [42,43]. A housing condition framework was applied by [44] in the slums of Dhaka, Bangladesh, based on four indicators (housing structure, tenure status, housing material, and room density), to determine the correlation between housing conditions and livelihood assets.

Each of the aforementioned approaches results in its own unique set of assumptions and interpretations, providing us with only a fragmented picture of life in slums. Despite their differences, however, these instructors are linked with one another. This paper presents a framework for understanding and studying slums from a combined approach of socio-economic and geo-physical contexts, building on previous research and being motivated by the need for a more holistic approach to studying slums. This approach helps to capture the variability in slum status by identifying developmental needs of the slum, as well as understanding the monitoring and analysis challenges associated with studying slums.

However, advances in remote-sensing technology, emergence of crowdsourced information on slums, and advancements in modelling enable one to better understand the complex nature of slums, and these advancements provide new opportunities to address the problem. With this framework in mind, this paper's main contribution is twofold: first, it identifies the most pressing issues surrounding slums based on current understandings, and second, it prescribes the treatment to be given for uplifting the slum conditions, which is what policymakers are looking for.

Jammu City in the Western Himalayas, known for its delicate ecosystem, has experienced rapid urbanization over the past 30 years, putting enormous strain on the city's social infrastructure [45]. The city dwellers' physical quality of life is impacted because the city's development body did not keep up with the rapid urbanization. Since Jammu has the maximum number of slums in Jammu and Kashmir and is going through the rapid development phase including the implementation of a smart city program. In order to overcome the worsening conditions of slums, this study has been carried out to identify the highly deteriorated slums in Jammu based on prioritization matrix analysis, with the incorporation of Management Information System (MIS) and Geographic Information System (GIS).

This research will allow officials to target the most dilapidated slum areas with a huge housing initiative funded by the Government of India's RAY plan with a focus towards improving residents' access to essential services.

## 2. Materials and Methods

Jammu, often known as the "City of Temples", is located in the south-western district of Jammu and Kashmir at a mean elevation of 327 m above sea level. Its coordinates are 32°43′58.79″ N latitude and 74°51′51.38″ E longitude. With its four Tehsils (Jammu, Akhnoor, Ranbir Singh Pura, and Bishnah), Jammu serves as the divisional headquarters and winter capital of the Jammu and Kashmir Union territory. Temperatures range from 4 °C to 47 °C and 1070 mm of rain falls annually here, indicating a subtropical to moist temperate climate [46]. The Himalayas to the north and the northern plains to the south surround Jammu City, which has a total size of 240 km². The city is enclosed on three sides by the Shivalik Range and on the fourth by the Trikuta Range. In general, the terrain in these regions is hilly, with slopes ranging from moderate to steep and linear ridges standing at a height of a few meters [47]. A strip of flat ground runs along the city's southern edge.

One can think of the entire district as being split in half by the Jammu–Chhamb road and the Jammu–Pathankot road. There are 152,406 people living in the Jammu district, while the urban area is home to another 657,314 people, as per the 2011 census. Jammu has a population density of 596 people per square kilometer, a population growth rate of 12.48 percent, a sex ratio of 871 females per 1000 males, and a literacy rate of 83.98 percent. Figure 1 is a basic map of Jammu City.

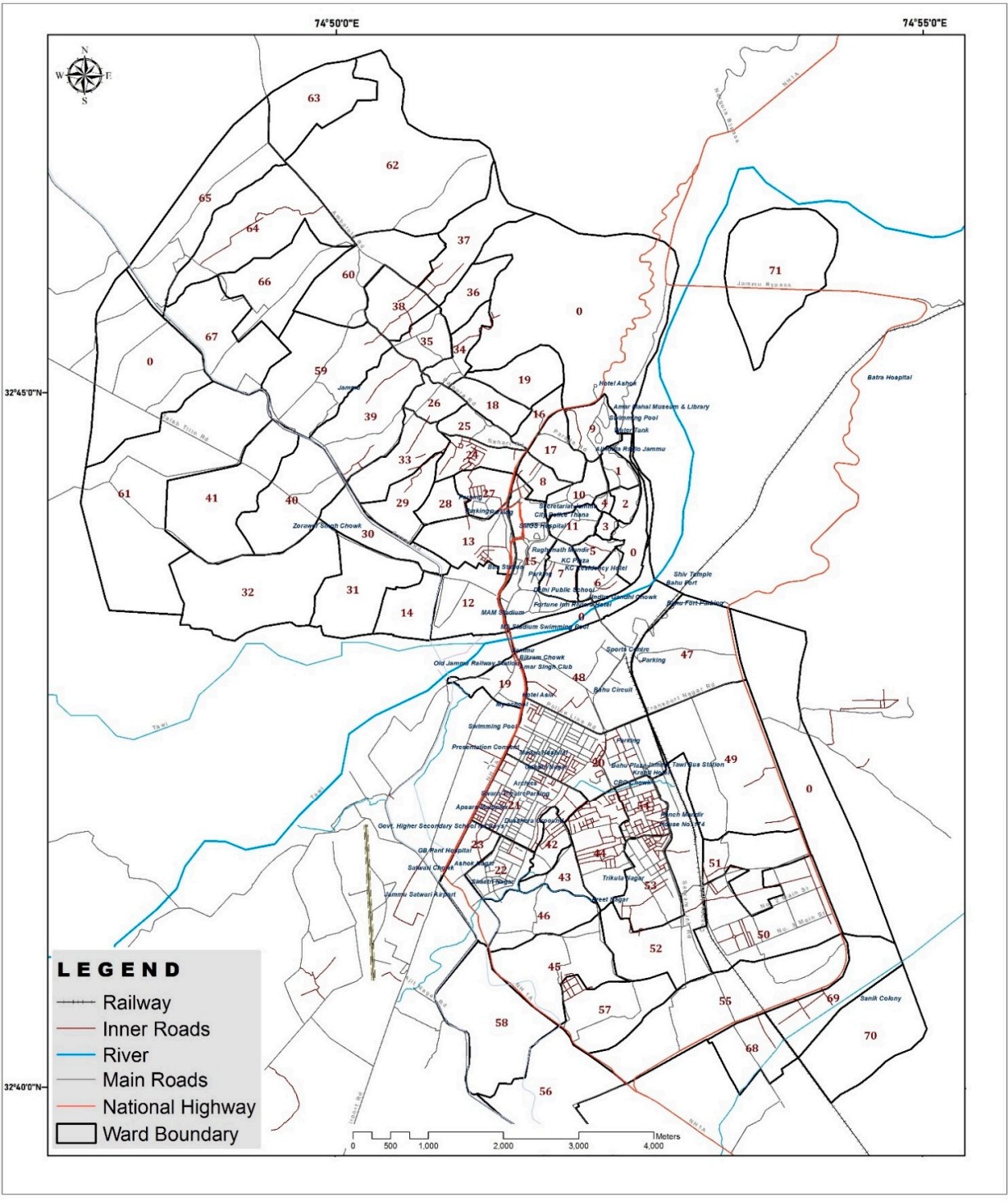

**Figure 1.** Base map of Jammu City prepared using municipal ward boundaries procured from Jammu Municipal Corporation.

Slum area surveys, maps, a MIS survey, and a matrix analysis form the basis of this study's conceptual framework. In this research, geo-physical and socio-economic considerations were central to the construction of a slum redevelopment strategy for Jammu City (Figure 2). The research considers a number of crucial physical criteria, including land use/land cover (LULC), road network, drainage, and current urban facilities. LULC is useful for pinpointing where a land-cover transition has occurred, and how rapidly it has occurred, because it is directly correlated with urbanization (see Figure 3) [48]. After georeferencing, the ward boundaries were drawn using Cartosat-1 satellite data with a spatial resolution of 2.5 m, and base layers were created using a map of municipal wards obtained from the National Remote Sensing Centre (NRSC) (Figure 3). It was not possible to accurately demarcate home borders from Cartosat-1 images, thus GPS coordinates were acquired in order to create a point layer in a geographic information system. Points such as these were clustered together to make ghetto walls.

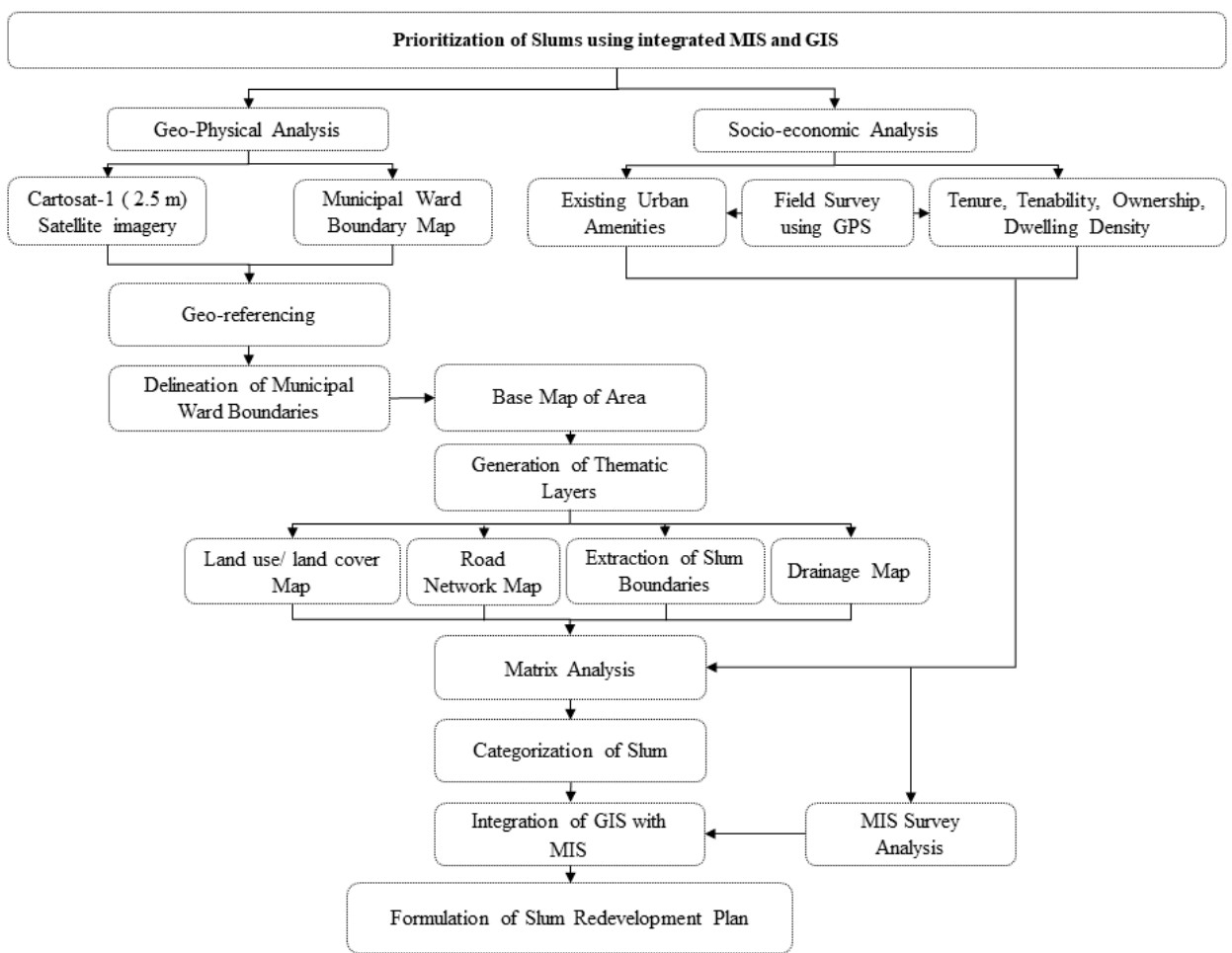

**Figure 2.** Flow chart of methodology.

The dataset collected from the field survey (tenure status, tenability, ownership, dwelling density) was integrated into the Management Information System (MIS) in order to link with the GIS household points collected during the survey of Jammu's slums for location, population, household size, and socio-economic profile. We considered infrastructure, tenure, and land value as key factors in developing the matrix. One of the most crucial aspects of the planning process is the tenure status of the slums, which is used to create a matrix.

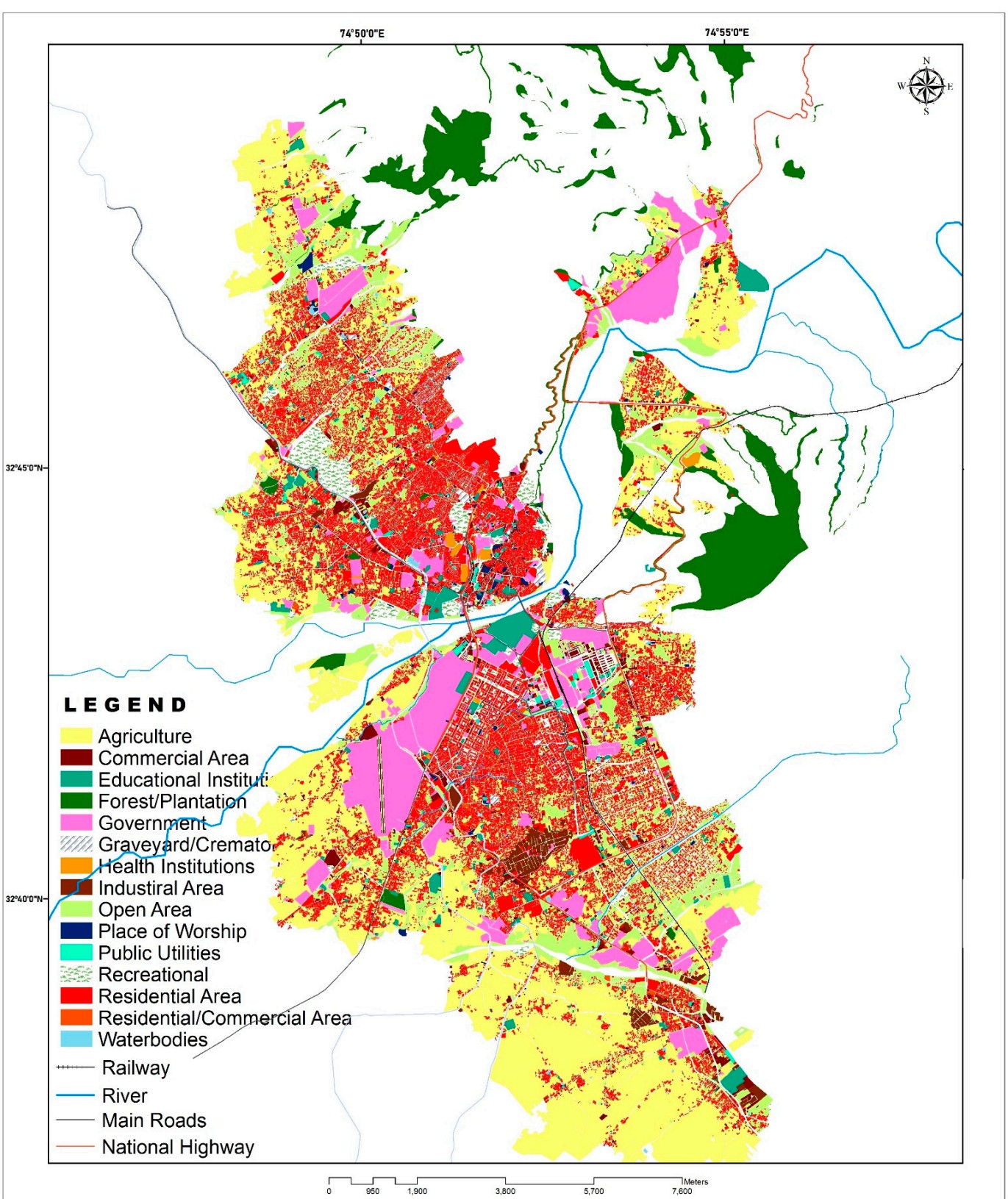

**Figure 3.** Land Use/Land Cover map of Jammu City prepared from Cartosat-1 satellite imagery overlaid on municipal ward boundaries of Jammu Municipal Corporation.

The slums that are not located on hazardous places and the land that is not planned for any important public infrastructure are suited for human habitation, hence it can be regularized in the same location as the official definition of "tenability" for slums under the Rajiv Awas Yojana (RAY). Using objective and verifiable factors (land ownership, land use, spatial position, health risk, etc.), all extant slum communities were recognised and designated as Tenable or Non-Tenable, depending on their level of sustainability. A matrix for classifying slums according to RAY standards, with categories including "Infrastructure", "Housing", and "Tenure Status", was developed to determine which areas are most in need of rapid improvement (Table 1). Sustainable development planning, according to Naess (2001) [49], necessitates integrating both specialised and generalised forms of knowledge. However, local residents' experimental knowledge of environmental quality could make up for the lack of expert knowledge, and a shift in residents' value priorities on development projects might be necessary for sustainable development. Keeping in view that the majority of community members did not have adequate education required to assign weightages, we conducted an exercise with community members to rank fundamental services/requirements in the order of importance and in their view and establish a consensus over the preference of identified solutions. An expanded method, as proposed in [50], is needed to assign relative importance to the various dimensions of slum improvement from the perspective of locals. A priority weightage method proposed by Barron and Barrett (1996) has been widely used [51]. The same approach was followed in our case as well, because it only asks participants to rank the listed objectives in their desired order. We subsequently calculated the average percentage based on those ranks, thus making the exercise accessible for the community. The most important attribute has the highest percent of priority weight, and the least important attribute has the lowest weight. The priority matrix (2 × 2 × 2) was used in this study for ranking the severity and prioritizing the slums into two classes i.e., high and low priority for intervention and as per the requirement of local government. The infrastructure metric considers factors such as the share of homes without running water, the share of homes with electricity, the share of homes with indoor plumbing, and the share of homes with toilets. The portion of households with unstable housing is included in the tenure section. In terms of housing, a higher percentage of Kutcha houses (houses made of mud and straw), as on priority weightage, are included than Semi–Pucca households (houses with fixed walls made of concrete material but roof made of hay). A total component score was calculated by summing the scores of each parameter, with each score representing the proportion of the above-mentioned components that are missing (Table 2). Table 1 details the seven criteria and three components that went into creating the 2 × 2 × 2 matrix. The ratings were calculated by giving each factor a certain weight based on what was learned from studies and community conversations.

**Table 1.** Infrastructure, Tenure, and Housing Deficiency Parameter.

| | | Parameter | Deficiency Score Range | |
|---|---|---|---|---|
| | | | 1 | 2 |
| 1. | | % of Household with Water Connection | 61–100 | 0–60 |
| 2. | Infrastructure | % of Pucca Roads | 61–100 | 0–60 |
| 3. | | % of Households with Electricity | 61–100 | 0–60 |
| 4. | | % of Households with Toilet facilities | 61–100 | 0–60 |
| 5. | **Tenure** | Insecure Tenure | 0–40 | 41–100 |
| 6. | **Housing** | % of Kutcha Housing (Structural Condition) | 0–40 | 41–100 |
| 7. | | % of Semi-Pucca Housing | 0–40 | 41–100 |

**Table 2.** Criteria for Percentage deficiency of Infrastructure, Tenure, and Housing on 2-point scale.

| Parameter | Total Score | Index |
|---|---|---|
| **Infrastructure Component** | ≤7 (High level of Infrastructure) | 1 |
| | >7 (Low level of Infrastructure) | 2 |
| **Tenure Component** | Secure Tenure (Good) | 1 |
| | Insecure Tenure (Poor) | 2 |
| **Housing Component** | 1–2 (Good) | 1 |
| | >2 (Poor) | 2 |

## 3. Results and Discussion

### 3.1. Slum Survey, Investigation, and Analysis

#### 3.1.1. Slum Locations

Jammu City is divided into 68 municipal wards. Out of 68 wards, only 23 wards consist of slums and fifteen Plan Areas have been designated where 15 slums exist. Out of 54 slums, six are notified in the core area and six are notified in the fringes (Figure 4). Eleven slums are non-notified in the core area and 31 slums are non-notified in the fringes. Ward 49 has the highest number of non-notified slums and ward 47 has the highest number of notified slums (Figure 5). Figure 6a,b show the distribution of notified and non-notified slum areas in Jammu City in satellite and map view, respectively.. Among the slum areas, Ward 47 has the largest percentage (9.08%) of the city's overall slum population, followed by Rajiv Colony Ambadkar Mohalla, Plan Area 02 (7.26%). Zorawar Singh Chowk, Ward 42 is the least-inhabited slum in terms of proportion of the total slum population.

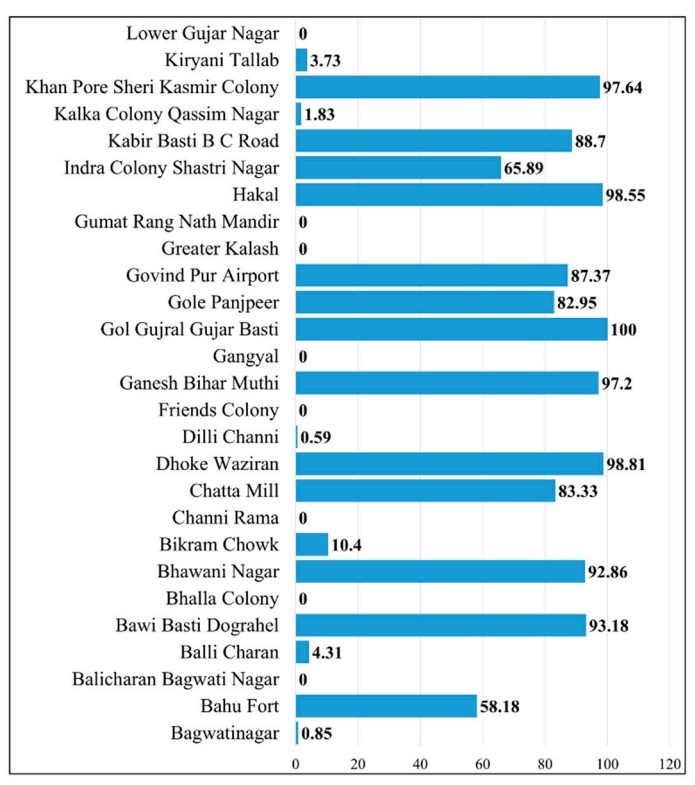
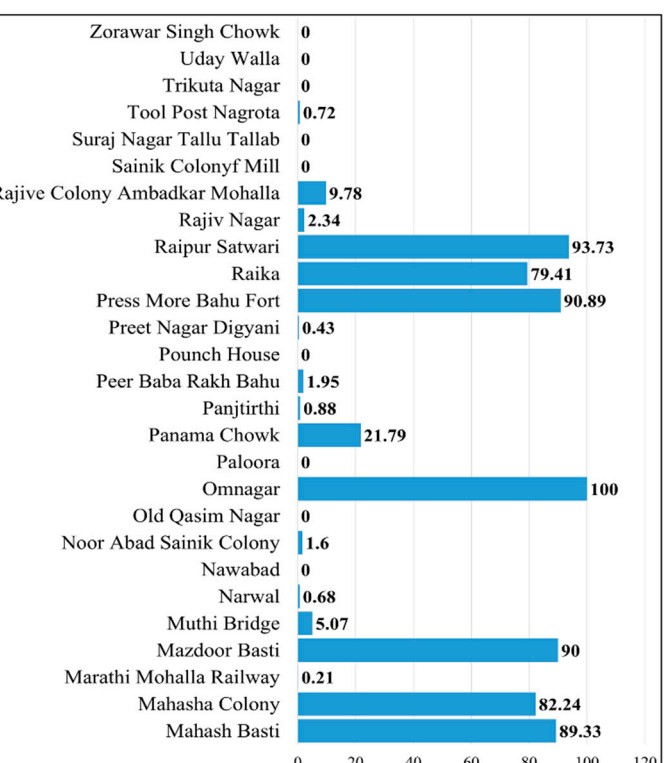

**Figure 4.** Percentage of Households having Possession certificate/Occupancy rights.

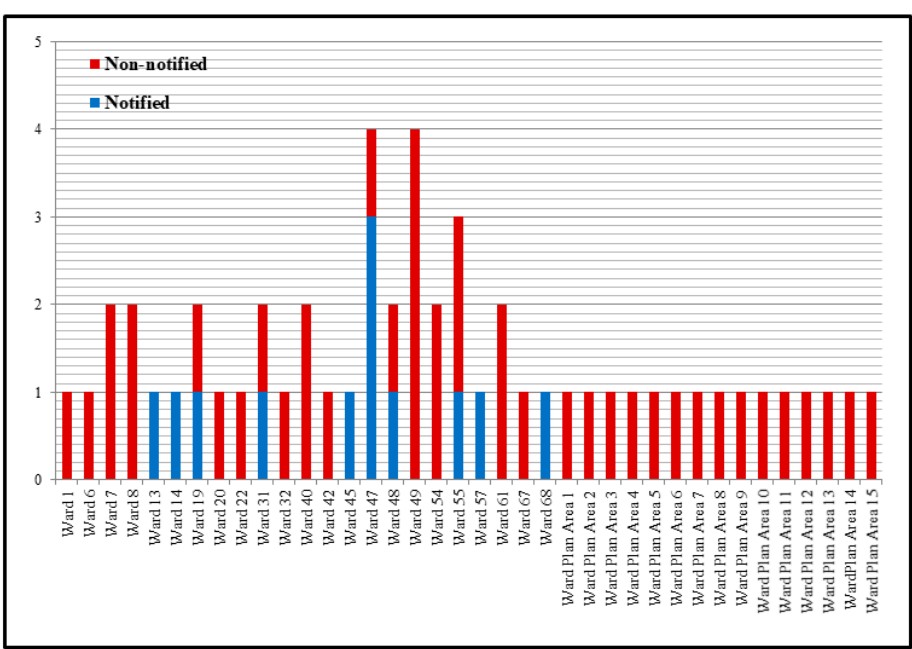

**Figure 5.** Ward-wise number of slums.

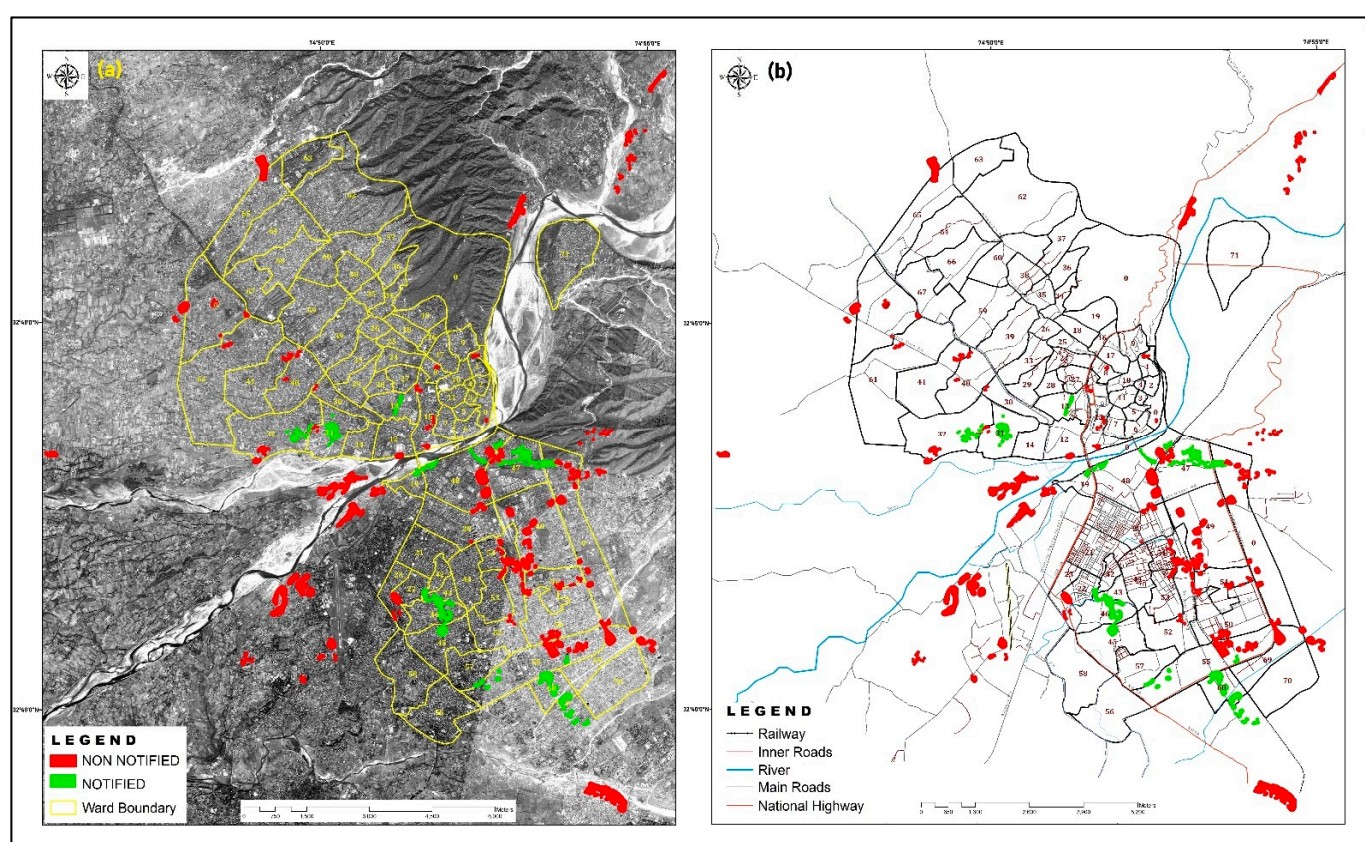

**Figure 6.** Showing distribution of notified and non-notified slum areas within ward boundaries of Jammu City (**a**) overlaid on cartosat data, (**b**) direct view.

Figure 4 displays detailed information about the tenancy of each Jammu slum. About one-quarter of the slum's 2952 residents (27.65 percent) have some sort of legal documentation proving their right to live there. About 43.82 percent of the population resides in rented dwellings; another 3000 people (28.10 percent) call encroached public property home;

another 27 people (0.25 percent) call encroached private land home; and just 14 people (0.13 percent) have the Patta.

From Appendix A it is clearly depicted that Ward 47 and Ward 49 have four slums each, Ward 55 has three slums, and Wards 8, 19, 31, 40, 48, 54, and 61 have two slums each. All other wards including the Plan Areas are individually inhabited by only one slum each. Rajiv Nagar of Ward 47 has the highest percentage of population at 9.08 % with 942 households followed by the slum 'Ambadkar Mohalla' Plan Area having 777 households comprising 7.26% of the total slum population of Jammu. Out of 54 slums across the wards enlisted in the above table 'Zorawar Singh Chowk' is the least populated with only 21 households and 101 souls. There are other slums such as 'Bhawani Nagar', 'Nawabad', and 'Raika' in Ward 31, 19, and Plan Area 04 having only 42, 43, and 34 households and populations of 154, 179, and 166, respectively. 'Old Qasim Nagar' and 'Bawi Basti Bograhel' in Wards 7 and 8 are also thinly populated with only 39 and 44 households and 185 and 188 inhabitants, respectively. The variation in the population and the number of households is enormous across the slums in the district; some are densely populated while others are moderately and some thinly populated. An analysis of the area surrounding the slums showed that 45 out of the 54 slums are inhabited in close proximity to the residential areas and commercial areas. In Pune, India, Ref. [52] also discovered that proximity to the city's central business district was a crucial factor in determining where slums were built.

Four slums are located in the commercial areas and the rest are in the vicinity of institutional or other localities. A majority of the slums, 20 of them, are located along the Nallah, fourteen slums are located in the vicinity of hazardous sites, five are located along river/waterbody banks, si near non-hazardous sites, five along railway lines, two along other drains, and only one each on river/waterbody beds and along the major transport establishment (Figure 7).

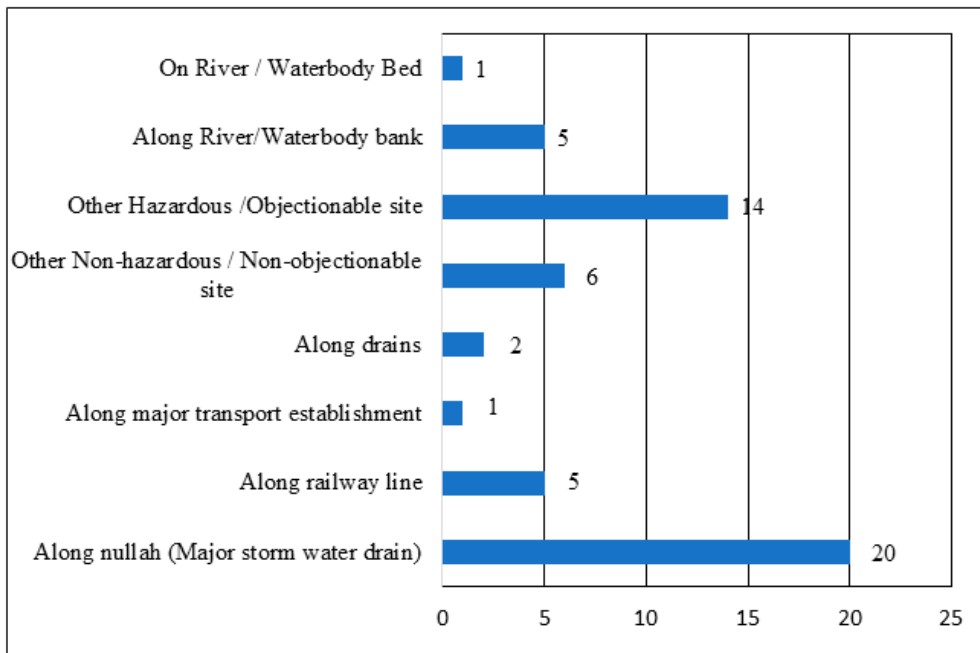

**Figure 7.** Physical Location Status of the Slums.

3.1.2. Slum Population

Appendix A Table A2 gives the population statistics of the surveyed slums, male–female population statistics, sex ratio, and family size. There are 10,674 households accommodating 46,838 persons in these slums. The community-wise percentage of each slum which reveals that community that lives in most of the slums is predominantly Hindu. However, some slums have a predominance of Muslims, such as Lower Gujjar Nagar, Khanpora Sheri Kashmir Colony, Raika, Balicharan Bhagwat Nagar, Gol Gujjar Basti, and

Kiryani Talab. There is only one Christian household each in Bawi Basti Dogral, Mazdoor Basti, Gole Panjpeer, Noorabad Sanik Colony, Khanpora Sheri Kashmir Colony, and Kiriyani Talab. There are, however, 11, six, five, four, and three Christian households in Rajiv Colony Ambedkar Mohalla, Mahesh Basti, Dhoki Waziran, Peer Baba Rakh Bahu, and Bahu Fort, respectively. There are 69 families of Sikhs in Govind pur Airport Slum out of the total 95 households.

The male–female population in all the slums shows an encouraging trend as far as females are concerned which is somewhat above the national and state figures. Thirteen out of 54 slums have a sex ratio above 1000, meaning that number of females is more for every thousand males. Except for Kabir Basti B C Road, Bagwati Nagar, Bikram Chowk, Suraj Nagar Talab Tillu, Bahu Fort, Uday walla, Ganesh Biahr Mutti, and Balicharan Bagwati, having sex ratios of 788, 844, 875, 809, 868, 795, 837, and 876, respectively, 46 out of 54 slums have a higher sex ratio compared to the state figure of 883. The average sex ratio in all the 54 slums is 945 which is very encouraging, far above the national and state level.

The number of persons for each household was analysed religion-wise. There are 8421 Hindu households, 2088 Muslim households, 129 Sikh households, and only 35 Christian households in all the 54 slums. The average Muslim family size is 4.74 and the Hindu family is 4.25. The average Christian family size is 4.82 and that of Sikhs size is 4.49. There are 622 households headed by women out of the total 10,674 households in all these slums.

### 3.1.3. Household Size

The number of households and the population density is maximum in Ward 47 having 1855 households accommodating 7907 slum dwellers in four slums. Ward 47 accounts for 16.92% of the total slum population of Jammu, followed by Ward 49 which accounts for 14.44% of the slum population with 1438 households carrying 6750 slum dwellers infour slums. The smallest ward is Ward 06 covering only 0.15% of the total slum population with 16 households sheltering 69 souls. The analysis of slum population provides an insight into the density of the slum pockets where more concentration of resources shall have to be deployed.

### 3.1.4. Socio-Economic Profile

One of the most crucial goals of the poll was to gauge the economic health of the region by looking at people's incomes. In total, there were 46,985 members, but only 10,678 were contributors (or 22.73 percent) according to the poll. One breadwinner per family is the typical figure. Casual workers made up the largest segment of this income bracket, accounting for 9498 people or 8.9% of the total. There were 745 individuals who were self-employed, accounting for 6.98% of the total earning members, whereas just 249 people were discovered to be with wages, accounting for around 2.333% of the total earning members. However, only 1.09 percent of the membership receives a regular salary.

### 3.2. MIS—Survey Analysis

The survey conducted in the slums of Jammu discussed in the preceding sections of the report was systematically uploaded into the Management Information System for online monitoring purposes. Each parameter in each household was captured by the questionnaire then organized and uploaded to have a methodical and orderly sequence of each variable. The Ministry of Housing and Urban Poverty Alleviation, National Building Organization, Government of India provided the website www.surveys.cgg.gov.in (accessed on 31 January 2021), which integrates the extensive socio-economic survey with the present physical living conditions in all of these 54 slums precisely conducted through an elaborate survey of all the households (Figure 8).

### 3.3. Matrix Analysis

Slum areas are given priority in the RAY guidelines by means of matrices. This would aid government officials in prioritizing initiatives in slums with the goal of addressing

the identified problems. Since resources are finite, prioritization is essential so that those resources are allocated where they will have the greatest impact. Matrices were developed to classify the slums based on factors such as the availability of essential services, the tenure of the residents, and the quality of the housing (Figure 9). Most of Jammu's slums are $1 \times 1 \times 2$ s or $1 \times 2 \times 2$ s.

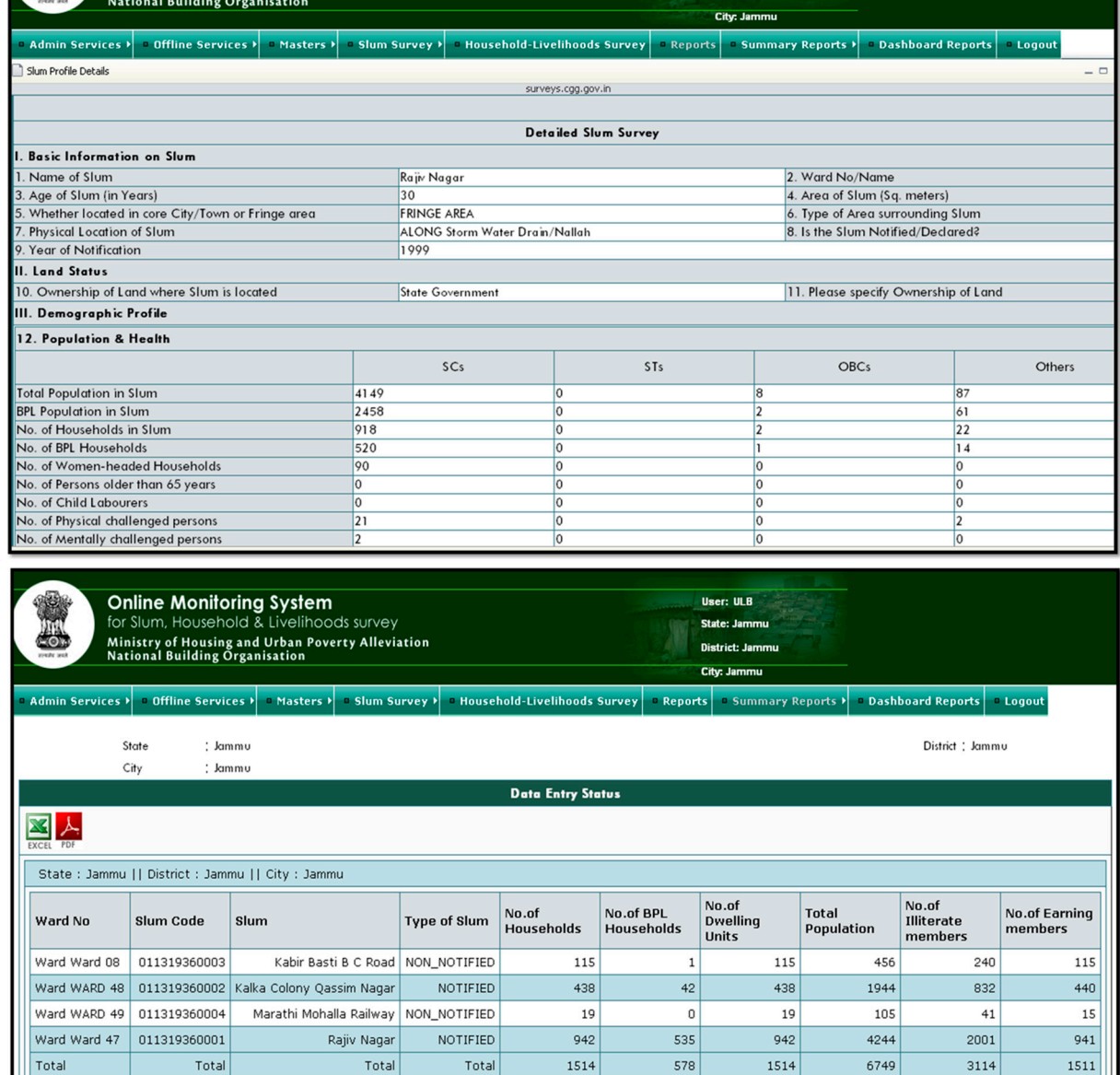

**Figure 8.** MIS interface (Household survey form, level report).

Each of the 54 shantytowns was ranked (represented by an element in a 222 matrix) according to their level of infrastructure decay, stability of tenure, and quality of housing based on the aforementioned criteria. If a slum received a final score of 1 for infrastructure, 1 for tenure, and 2 for housing, it would be ranked as $1 \times 1 \times 2$. Final grades of excellence or failure were assigned to each matrix category. In the above matrix, 21 shantytowns are located in the $1 \times 1 \times 2$ box, while 33 others are in the $1 \times 2 \times 2$ box. The former describes slums with adequate or above-average infrastructure, stable tenure, and subpar living

quarters. In the latter situation, infrastructure is adequate, but tenants face instability in their leases and subpar living circumstances.

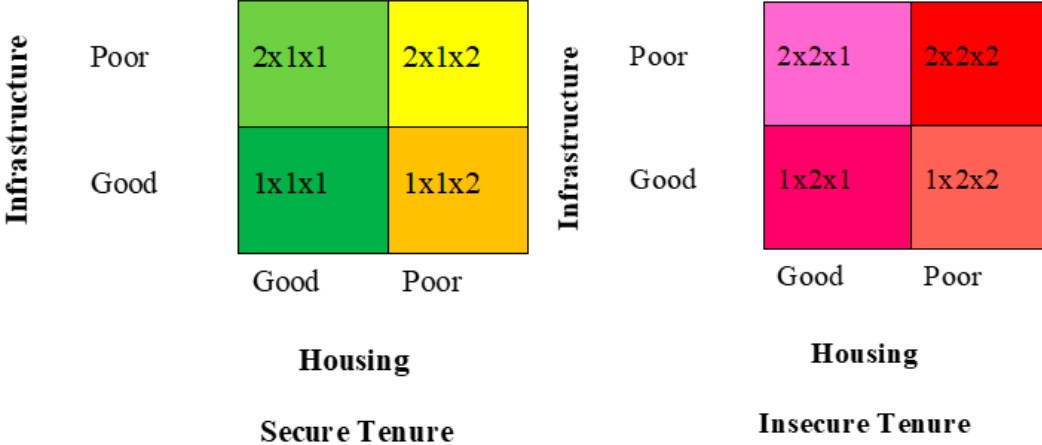

**Figure 9.** Matrix Categorization.

From the matrix analysis, out of 54 slums identified in Jammu City, 21 slums are placed in a $1 \times 1 \times 2$ matrix having good or fair infrastructure, secure tenure, but poor housing conditions. A total of 33 slums fall in $1 \times 2 \times 2$ with infrastructure fairly good, tenure insecure, and housing conditions poor. Figure 10a depicts the distribution of slums based on tenure status and tenability. Figure 10b shows the status of land ownership and dwelling density among slums of Jammu City.

The 21 slums have 3569 households, which accounts for 33.43% of the total number of households in the slums. Out of a total slum population of 46,985 persons, the population residing in these slums is 14,994, which accounts 31.92% of the total slum population. Tenure-wise two slums have 100% and eight have more than 90% secure tenure. Six range between 80 % and 90%, two range between 60% and 80%, and one has only 3% secure households. One slum, Poonch House, had no households with secure tenure. The overall housing condition is very poor as 72.56% and 24.45% are Kutcha houses and Semi-Pucca houses, respectively. Kabir Basti BC road, Poonch House, and Gol Gujral Gujjar Basti are the slums with 100% Kutcha houses. Only 33.84% of households had in-house water connections, 14.17% had Pucca roads (permanent roads made of concrete material), but 99.92% had electrical connections. Again, only 0.45% of the households had in-house toilet facilities. A lack of public participation in the management of public facilities has been cited as a major contributor to the poor state of public restrooms in Mumbai's slum area [53,54].

The population residing in these 33 slums is 31,844 which accounts for 67.77% of the total slum population. The overall housing condition is very poor with Kutcha (92.14%) and semi-Pucca (7.59%) houses. Only 4.78% of houses have in-house water connection. Only 31.48% of the households have access to Pucca roads and 0.64% have the toilet facility within the premises. However, 98.71% of the households in these slums are connected with electricity. According to Ref. [55], slum areas are home to a number of economically fruitful enterprises, making reliable access to electricity and clean water especially important. Comparable results have been reported in Refs. [56,57], where the authors show that slum residents heavily weigh the availability of local services when deciding where to settle down.

Tenure-wise, 96.25% of the households have an insecure tenure. Lower Gujjar Nagar, Gumat Ragunath Mandir, Old Qasim Nagar, Nawabad, Suraj Nagar Tallu Talab, Zorawar Singh Chowk, Bhalla Colony, Channi Rama, Friends Colony, Trikuta Nagar, Sainik Colony, Gangayal, Paloora, Uday Walla, and Greater Kalash are slums with 100% insecure tenure. Figure 11a,b presents the percentage of households under matrix parameters for $1 \times 1 \times 2$ and $1 \times 2 \times 2$ matrices, respectively, with detailed data in Appendix A Tables A3 and A4. Figure 12 presents the distribution of slums as per matrix profile. Planning for new

migrants to the city and preventing the construction of new slums is an important part of slum upgrading.

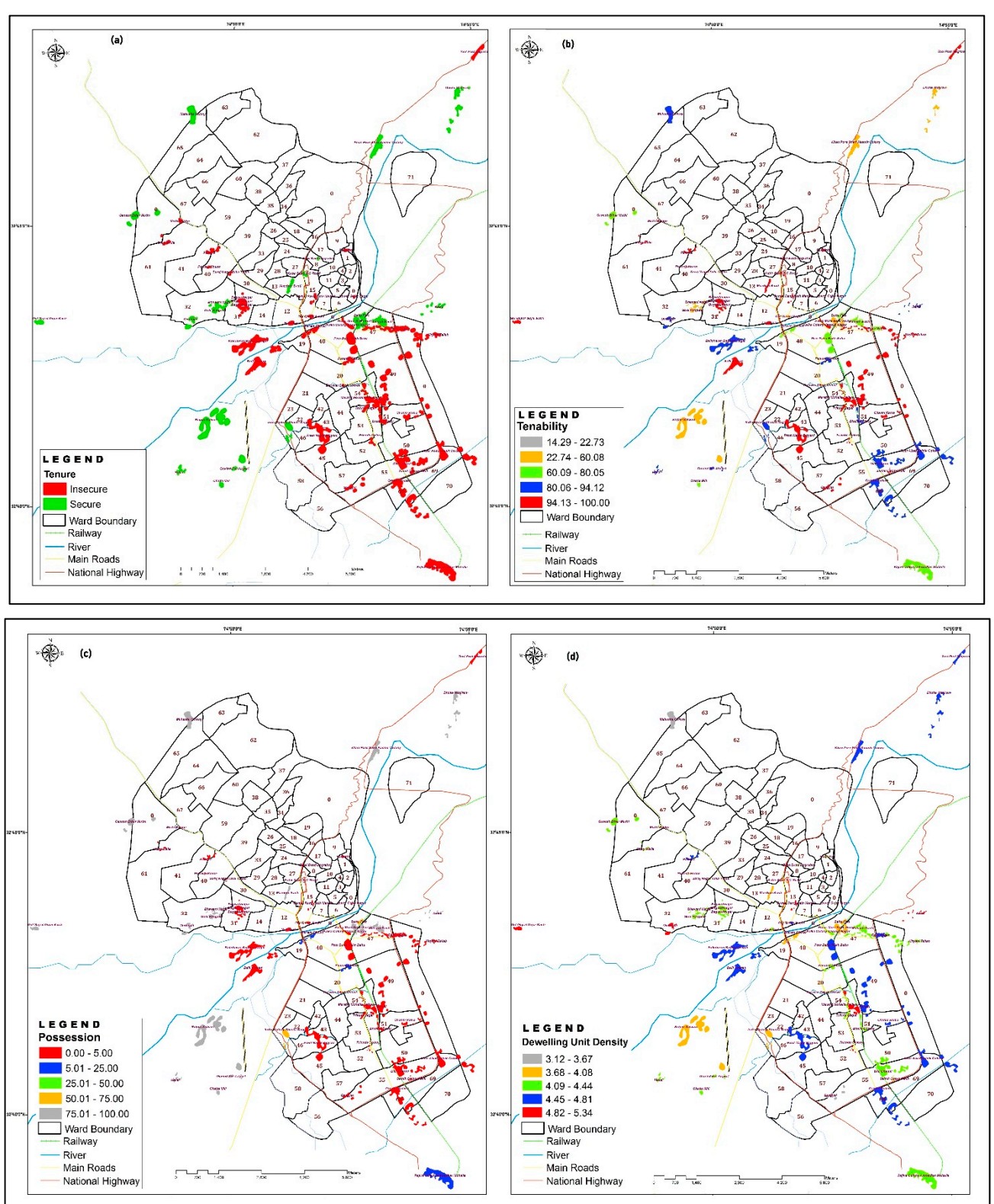

**Figure 10.** Map showing slum status based on (**a**) tenure, (**b**) tenability, (**c**) land ownership, and (**d**) dwelling-unit density.

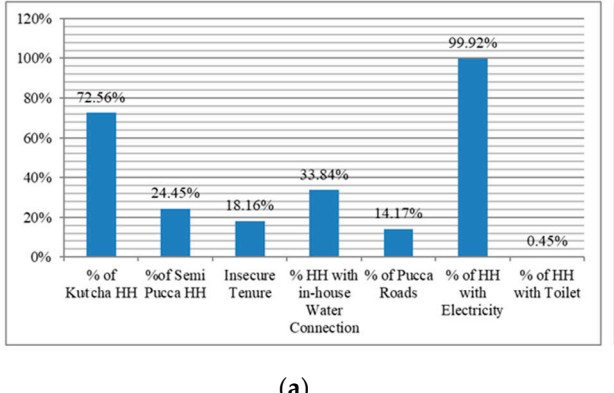
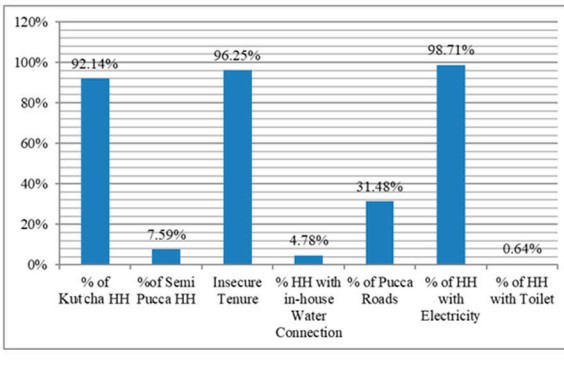

<div align="center">(<b>a</b>)　　　　　　　　　　　　　　　　　　　　　　　(<b>b</b>)</div>

**Figure 11.** (**a**) percentage of amenities in $1 \times 1 \times 2$ matrix profile (**b**) percentage of amenities in $1 \times 2 \times 2$ matrix profile.

### 3.4. Slum Development, Prevention Strategy, and Implementation Plan

3.4.1. Prevention Strategy and Development of Slum

To make Jammu slum free, a Slum Development, Prevention Strategy, and Implementation Plan has been devised taking into cognizance the broad outlines put forth by the Government of India through the guidelines of RAY. The Slum Prevention Strategy, which aims to stop the spread of slums, is based on the following six guiding principles:

1.　Fix things up where they are instead of forcing folks to uproot.
2.　Give the poor secure ownership of all developable land and help them invest in their own housing and infrastructure improvements.
3.　Residents should hold on to their investments for as long as possible to ease the strain on JMC's infrastructure and allow for faster expansion.
4.　Involve people in service design, development, delivery, and upkeep.
5.　Provide families with legal access to service and the opportunity to pay for their own connections, monthly fees, etc.
6.　Affordability-based increases in slum residents' own housing-upgrade investments lead to a reduction in the total subsidy.

Based on the above-mentioned key principles, Jammu Municipal Corporation (JMC) may undertake the Slum Upgradation in Jammu City for present and future slums as given in Figure 13.

3.4.2. Development Plan

Analysis of the slums places 54 of them in the $1 \times 1 \times 2$ group and 23 in the $1 \times 2 \times 2$ group. Thus, there are two types of people living in the Jammu slums. Upgrading slums entails not just addressing the slums already there, but also preparing for the arrival of new residents to the city. As a result, the towns have been divided into three clusters for easier upgrading (Table 3). The Government of India has issued regulations mandating a certain course of action for the development of the designated clusters. Tables 4 and 5 detail the various clusters and their corresponding development choices.

**Table 3.** Classification of the settlements.

| Classification of Settlements | | |
|---|---|---|
| **Cluster 1** | **Cluster 2** | **Cluster 3** |
| $1 \times 1 \times 2$ | $1 \times 2 \times 2$ | |
| Secure Tenure | Insecure Tenure | |
| Good Infrastructure | Good Infrastructure | New Migrants |
| Poor Housing conditions | Poor Housing conditions | |

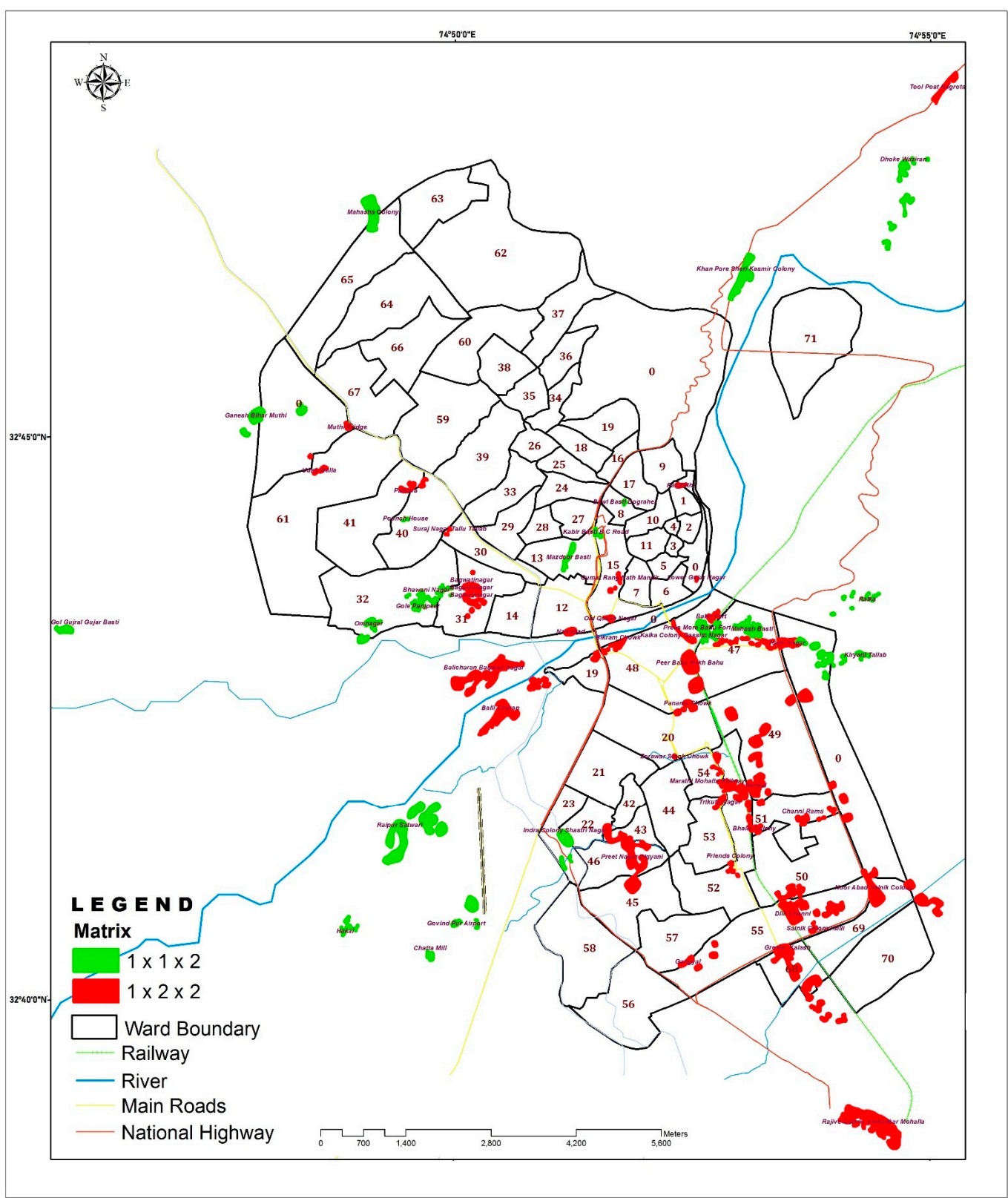

**Figure 12.** Map showing different slums as per Matrix Profiles.

The future directions of all these communities have been planned out. The Development options shall include in-place redevelopment through land sharing and PPPs that include a sales component, in-place redevelopment through densification that allows for $1 \times 1 \times 2$ and $1 \times 2 \times 2$ uses, and in-place redevelopment through densification that allows for $1 \times 1 \times 2$ uses. In any case, JMC has the option of relocating, which it may or may not pursue depending on the price and accessibility of suitable new real estate.

**Prevention Strategy and Development of Slum**

Slum Up-gradation Plan

Inclusion

Provisioning Secure Land Tenure

In-situ Upgradation in Tenable Sites

Providing Housing for All

Minimizing Relocation

Resource Mobilization

Access to Social Services

Improving Access to Micro-finance

Equitable Norms and Standards for Municipal Services

Community Participation and Organization

Promoting Sustainable Livelihood

Institutional Arrangements and Partnerships

Monitoring and Impact Assessment

Focus on Enabling Policy and Reform Framework

**Figure 13.** Slum Upgradation Plan for Jammu City.

**Table 4.** Selection of Development Option ($1 \times 1 \times 2$).

| Deficiency Index | Slum Deficiency and Vulnerability | Land Ownership | Land Value | Dwelling Unit's Density | Development Option |
|---|---|---|---|---|---|
| $1 \times 1 \times 2$ | Secure Tenure with poor Housing and good Infrastructure | Public Municipal or State | High | High/Medium | Property-based partnership (PPP) for on-site renovation (community dwellings). |
| | | | | Low | On-site renovation involving a public-private partnership (PPP) and a sale of renovated properties |
| | | | Low | High/Medium | Assistance from a government organisation and financing for on-site renovations |
| | | | | Low | Densification and in-place renovation with credit for the work done |
| | | Private | High | High/Medium | Promotional Floor Space Index (FSI) or Transfer of Development Rights (TDR) for Construction Within Existing Areas |
| | | | | Low | Conversion of existing structures on a site that is jointly owned with the landowner |
| | | | Low | High/Medium | Redevelopment in its present location; in situ redevelopment; transferable development rights |
| | | | | Low | Government-led redevelopment taking place on-site, with landowner input |

Residents of an area that is undergoing in situ rehabilitation have the option of remaining in their homes during the construction phase so that they can remain near their present employment places and social networks. A successful cross-subsidy model can be executed through redevelopment of areas with existing slum settlements, which depends on socio-economic demands, local land prices, and the overall context of the place. Therefore, it is crucial to conduct a socio-economic analysis of the surrounding area before making any decisions. It is possible to reconstruct places in situ, especially those with a medium or high population density.

**Table 5.** Selection of Development Option ($1 \times 2 \times 2$).

| Deficiency Index | Slum deficiency and Vulnerability | Land Ownership | Land Value | Dwelling Unit's Density | Development Option |
|---|---|---|---|---|---|
| $1 \times 2 \times 2$ | Insecure Tenure with poor Housing and good Infrastructure | Public Municipal or State | High | High/Medium | Public sector in situ incentive-based FSI/PPP redevelopment |
| | | | | Low | Densification and in-place redevelopment, with the potential for monetary gain |
| | | | Low | High/Medium | Leases are made permanent, and public funds are used for renovations and regular upkeep. |
| | | | | Low | Densification and redevelopment in place. |
| | | Private | High | High/Medium | Redevelopment in place with FSI or TDR incentives |
| | | | | Low | Mixed-use redevelopment on existing land |
| | | | Low | High/Medium | Conversion of Existing Structures to New Uses, or "In Situ" purchase/acquisition. |
| | | | | Low | Distribution of Existing Land for Redevelopment Purposes |

The process of improving and expanding slum areas should be carried out in stages. The total number of homes in Jammu's 26 slums is 13,498, therefore the city's reconstruction effort can be split into two parts. According to the results of the study, 54 of Jammu's slums are located in Cluster 1, while 23 are in Cluster 2. All of Jammu's shantytowns are Tenable, thus building there is a great option. Clusters 1 and 2 will undergo renovations in Stage 1, while Cluster 3 will do so in Stage 2. Table 6 provides a timeline of both stages.

**Table 6.** Phase implementation.

| | Phases | | |
|---|---|---|---|
| Criteria | Cluster 1 | Cluster 2 | Cluster 3 |
| Index | $1 \times 1 \times 2$ | $1 \times 2 \times 2$ | New Migrants |
| Phase | Phase 1 | Phase 1 | Phase 2 |
| Time line | Year 1 to 3 | Year 1 to 3 | Year 3–6 |

3.4.3. Implementation Plan

It will take a well-thought-out implementation plan, phasing, timeline, and funding approach to improve the city's slums. However, slum data are not consistent and may evolve with shifting demographics, economies, geographies, and/or policies. While the specifics of the intervention may change over time, the strategies for putting them into action will not. The plan, which was developed using data collected through field surveys and mapping of existing slum information and their reach to basic amenities in the city, requires the measures shown in Figure 14 to be taken.

## Implementation Plan

| Implementing slum redevelopment |
| --- |
| Data on slums, land ownership and services |
| Development options |
| Preparing a phased action plan for upgrading |
| Planning for affordable housing for all |
| Estimated housing demand |
| Housing for new migrants |
| In-situ slum redevelopment |
| Provisioning municipal services |
| Notification and denotification of settlement |
| Access to social services |
| Convergence for improved education outcomes |

**Figure 14.** Implementation Plan.

### 4. Conclusions

Jammu City, like many others in India, has been troubled by a housing shortage and the expansion of slums as a result of urbanization. Slum revitalization, being the primary low-cost housing alternative for the urban poor, necessitates a more in-depth diagnostic examination. The policy makers have been disconnected from the actual needs of the slum residents due to the lack of involvement of the direct stakeholders. Residents of slums have evolved to the sociocultural environment of slums, and any developmental endeavor that disrupts this environment could have a negative influence on their well-being. Individual tenure legalization is the first and best potential solution for a slum to be rehabilitated. Understanding the capabilities, options, and willingness of slum inhabitants is essential for effective slum redevelopment and urban environment management, as is the strong commitment of the administration to establish and maintain a conducive atmosphere. For slums to be successfully revitalized and improved in the future, their management strategy must take into account all viable options, regardless of how novel or unconventional they may be.

This paper presented a comprehensive framework for the selection of slum redevelopment and upgradation strategies. The main objectives of this system were to identify slum areas in Jammu City for the prioritization and formulation of a slum redevelopment plan. The framework consists of two main phases, including (1) framework structuring (generating input data) and (2) framework implementation (optimizing intervention priorities). Improvements to squatter areas should be carried out in accordance with a sensible plan, phased over a reasonable amount of time, and funded in an approachable manner. The steps of the implementation plan include: Community Participation, Promoting Sustainable Livelihoods, Estimated Housing demand for existing residents and new migrants, Affordable Housing for All, Provisioning of Municipal, Health, Educational, and Social services, Notification and De-notification of Settlements, and Community Based Monitoring. In

addition, ecologically sensitive and environmentally hazardous sites should be considered untenable as habitation.

The foremost contribution of this research can be concluded in two main points. First, it recognized the most persistent problems surrounding slums based on current understanding. Second, it proposed the treatment to be given for uplifting the slum conditions, which is what policymakers are looking for. Although this study provides promising initial evidence on the informal settlements and their conditions in Jammu City, there is a lack of some important aspects which can be taken into consideration in future to overcome the limitations of this study. Rather than simply measuring the basic infrastructure and amenities, the analysis can be performed at a broader scale incorporating ladders that were recently proposed by the WHO/UNICEF Joint Monitoring Programme to further differentiate between households on the basis of levels of service quality.

**Author Contributions:** Conceptualization, M.F. and F.M.; methodology, M.F. and G.M.; software, M.F., S.K.S. and S.K.; validation, M.F. and P.K.; formal analysis, M.F.; investigation, M.F., A.G., D.S. and R.A.; writing—original draft preparation, M.F., F.M., G.M., S.K.S., S.K., P.K. and R.A.; writing—review and editing, M.F., F.M., G.M., S.K.S., S.K., A.G., P.K., D.S. and R.A.; visualization, M.F., F.M., S.K. and S.K.S. All authors have read and agreed to the published version of the manuscript.

**Funding:** This research was partially supported by Kakenhi Kiban-C (21K05664) and Kajima Foundation grants.

**Data Availability Statement:** Not applicable.

**Acknowledgments:** We are thankful to the three anonymous reviewers for their comments, that has greatly improved the quality of this manuscript. This research was partially supported by Kakenhi Kiban-C (21K05664) and Kajima Foundation grants.

**Conflicts of Interest:** The authors declare no conflict of interest.

## Appendix A

**Table A1.** Locations of identified slums.

| Ward No. | Slum Name | No. of Households | Slum Population | Share in Slum Population (%) |
|---|---|---|---|---|
| Ward 01 | Panjtirthi | 113 | 485 | 1.04 |
| Ward 06 | Lower Gujar Nagar | 16 | 69 | 0.15 |
| Ward 07 | Gumat Rang Nath Mandir | 61 | 326 | 0.70 |
| Ward 07 | Old Qasim Nagar | 39 | 185 | 0.40 |
| Ward 08 | Bawi Basti Dograhel | 44 | 188 | 0.40 |
| Ward 08 | Kabir Basti B C Road | 115 | 456 | 0.98 |
| Ward 13 | Mazdoor Basti | 60 | 241 | 0.52 |
| Ward 14 | Bagwatinagar | 354 | 1468 | 3.14 |
| Ward 19 | Bikram Chowk | 202 | 825 | 1.77 |
| Ward 19 | Nawabad | 43 | 179 | 0.38 |
| Ward 20 | Panama Chowk | 78 | 329 | 0.70 |
| Ward 22 | Indra Colony Shastri Nagar | 129 | 506 | 1.08 |
| Ward 31 | Bhawani Nagar | 42 | 154 | 0.33 |
| Ward 31 | Gole Panjpeer | 258 | 1119 | 2.39 |
| Ward 32 | Omnagar | 50 | 257 | 0.55 |
| Ward 40 | Pounch House | 59 | 241 | 0.52 |

**Table A1.** *Cont.*

| Ward No. | Slum Name | No. of Households | Slum Population | Share in Slum Population (%) |
|---|---|---|---|---|
| Ward 40 | Suraj Nagar Tallu Tallab | 79 | 340 | 0.73 |
| Ward 42 | Zorawar Singh Chowk | 21 | 101 | 0.22 |
| Ward 45 | Preet Nagar Digyani | 470 | 2246 | 4.81 |
| Ward 47 | Bahu Fort | 110 | 424 | 0.91 |
| Ward 47 | Mahash Basti | 375 | 1591 | 3.40 |
| Ward 47 | Press More Bahu Fort | 428 | 1648 | 3.53 |
| Ward 47 | Rajiv Nagar | 942 | 4244 | 9.08 |
| Ward 48 | Kalka Colony Qassim Nagar | 438 | 1944 | 4.16 |
| Ward 48 | Peer Baba Rakh Bahu | 154 | 703 | 1.50 |
| Ward 49 | Bhalla Colony | 173 | 752 | 1.61 |
| Ward 49 | Channi Rama | 191 | 879 | 1.88 |
| Ward 49 | Marathi Mohalla Railway | 482 | 2360 | 5.05 |
| Ward 49 | Narwal | 592 | 2759 | 5.90 |
| Ward 54 | Friends Colony | 89 | 442 | 0.95 |
| Ward 54 | Trikuta Nagar | 54 | 228 | 0.49 |
| Ward 55 | Dilli Channi | 338 | 1482 | 3.17 |
| Ward 55 | Noor Abad Sainik Colony | 125 | 586 | 1.25 |
| Ward 55 | Sainik Colonyf Mill | 86 | 339 | 0.73 |
| Ward 57 | Gangyal | 116 | 362 | 0.77 |
| Ward 61 | Paloora | 61 | 285 | 0.61 |
| Ward 61 | Uday Walla | 50 | 219 | 0.47 |
| Ward 67 | Muthi Bridge | 138 | 606 | 1.30 |
| Ward 68 | Greater Kalash | 183 | 835 | 1.79 |
| Plan Area 01 | Balli Charan | 255 | 1162 | 2.49 |
| Plan Area 02 | Rajive Colony Ambadkar Mohalla | 777 | 3393 | 7.26 |
| Plan Area 03 | Khan Pore Sheri Kasmir Colony | 508 | 2332 | 4.99 |
| Plan Area 04 | Raika | 34 | 166 | 0.36 |
| Plan Area 05 | Dhoke Waziran | 84 | 388 | 0.83 |
| Plan Area 06 | Ganesh Bihar Muthi | 143 | 621 | 1.33 |
| Plan Area 07 | Raipur Satwari | 287 | 1159 | 2.48 |
| Plan Area 08 | Balicharan Bagwati Nagar | 136 | 634 | 1.36 |
| Plan Area 09 | Mahasha Colony | 152 | 545 | 1.17 |
| Plan Area 10 | Gol Gujral Gujar Basti | 23 | 121 | 0.26 |
| Plan Area 11 | Hakal | 69 | 292 | 0.62 |
| Plan Area 12 | Kiryani Tallab | 536 | 2248 | 4.81 |
| Plan Area 13 | Tool Post Nagrota | 139 | 555 | 1.19 |
| Plan Area 14 | Govind Pur Airport | 95 | 398 | 0.85 |
| Plan Area 15 | Chatta Mill | 78 | 323 | 0.69 |
| Total | | 10,674 | 46,740 | 100 |

**Table A2.** Population of Surveyed Slums.

| Slums | Panjtirthi | Lower Gujar Nagar | Gumat Rang Nath Mandir | Old Qasim Nagar | Bawi Basti Dograhel | Kabir Basti B C Road | Mazdoor Basti | Bagwati nagar | Bikram Chowk | Nawabad | Panama Chowk | Indra Colony Shastri Nagar | Bhawani Nagar | Gole Panjpeer |
|---|---|---|---|---|---|---|---|---|---|---|---|---|---|---|
| Households | 113 | 16 | 61 | 39 | 44 | 115 | 60 | 354 | 202 | 43 | 78 | 129 | 42 | 258 |
| *% Hindu* | 91.15 | 0.00 | 100.00 | 92.31 | 97.73 | 96.52 | 95.00 | 94.07 | 75.74 | 95.35 | 38.46 | 96.12 | 100.00 | 91.47 |
| *% Muslim* | 8.85 | 100.00 | 0.00 | 2.56 | 0.00 | 3.48 | 1.67 | 5.93 | 24.26 | 4.65 | 61.54 | 3.88 | 0.00 | 6.98 |
| *% Christian* | 0.00 | 0.00 | 0.00 | 0.00 | 2.27 | 0.00 | 1.67 | 0.00 | 0.00 | 0.00 | 0.00 | 0.00 | 0.00 | 0.39 |
| *% Sikh* | 0.00 | 0.00 | 0.00 | 5.13 | 0.00 | 0.00 | 1.67 | 0.00 | 0.00 | 0.00 | 0.00 | 0.00 | 0.00 | 1.16 |
| *WHH* | 4 | 1 | 1 | 1 | 4 | 6 | 6 | 11 | 20 | 2 | 7 | 19 | 3 | 14 |
| **Persons** | | | | | | | | | | | | | | |
| *Total* | 485 | 69 | 326 | 185 | 188 | 456 | 241 | 1468 | 825 | 179 | 329 | 506 | 154 | 1119 |
| *Male* | 257 | 34 | 165 | 96 | 95 | 255 | 126 | 796 | 440 | 92 | 160 | 253 | 80 | 584 |
| *Female* | 228 | 35 | 161 | 89 | 93 | 201 | 115 | 672 | 385 | 87 | 169 | 253 | 74 | 535 |
| *Sex Ratio* | 887 | 1029 | 976 | 927 | 979 | 788 | 913 | 844 | 875 | 946 | 1056 | 1000 | 925 | 916 |
| **Family Size** | | | | | | | | | | | | | | |
| *Total* | 4.29 | 4.31 | 5.34 | 4.74 | 4.27 | 3.97 | 4.02 | 4.15 | 4.08 | 4.16 | 4.22 | 3.92 | 3.67 | 4.34 |
| *Hindu* | 4.21 | 0.00 | 5.34 | 4.81 | 4.28 | 3.97 | 3.98 | 4.10 | 4.09 | 4.07 | 4.03 | 3.85 | 3.67 | 4.33 |
| *Muslim* | 5.10 | 4.31 | 0.00 | 6.00 | 0.00 | 3.75 | 2.00 | 4.95 | 4.06 | 6.00 | 4.33 | 5.80 | 0.00 | 4.17 |
| *Christian* | 0.00 | 0.00 | 0.00 | 0.00 | 4.00 | 0.00 | 5.00 | 0.00 | 0.00 | 0.00 | 0.00 | 0.00 | 0.00 | 5.00 |
| *Sikh* | 0.00 | 0.00 | 0.00 | 3.00 | 0.00 | 0.00 | 7.00 | 0.00 | 0.00 | 0.00 | 0.00 | 0.00 | 0.00 | 5.33 |

| Slums | Omnagar | Pounch House | Suraj Nagar Tallu Tallab | Zorawar Singh Chowk | Preet Nagar Digyani | Bahu Fort | Mahash Basti | Press More Bahu Fort | Rajiv Nagar | Kalka Colony Qassim Nagar | Peer Baba Rakh Bahu | Bhalla Colony | Channi Rama | Marathi Mohalla Railway |
|---|---|---|---|---|---|---|---|---|---|---|---|---|---|---|
| Households | 50 | 59 | 79 | 21 | 470 | 110 | 375 | 428 | 942 | 438 | 154 | 173 | 191 | 482 |
| *% Hindu* | 52.00 | 98.31 | 100.00 | 95.24 | 98.30 | 61.82 | 91.47 | 98.60 | 97.88 | 85.16 | 51.30 | 97.69 | 98.95 | 89.83 |
| *% Muslim* | 46.00 | 1.69 | 0.00 | 4.76 | 0.43 | 35.45 | 6.13 | 0.70 | 2.12 | 14.61 | 44.16 | 2.31 | 1.05 | 10.17 |
| *% Christian* | 0.00 | 0.00 | 0.00 | 0.00 | 0.00 | 2.73 | 1.60 | 0.00 | 0.00 | 0.00 | 2.60 | 0.00 | 0.00 | 0.00 |
| *% Sikh* | 2.00 | 0.00 | 0.00 | 0.00 | 1.28 | 0.00 | 0.80 | 0.70 | 0.00 | 0.23 | 1.95 | 0.00 | 0.00 | 0.00 |

**Table A2.** *Cont.*

| Slums | Panjtirthi | Lower Gujar Nagar | Gumat Rang Nath Mandir | Old Qasim Nagar | Bawi Basti Dograhel | Kabir Basti B C Road | Mazdoor Basti | Bagwati nagar | Bikram Chowk | Nawabad | Panama Chowk | Indra Colony Shastri Nagar | Bhawani Nagar | Gole Panjpeer |
|---|---|---|---|---|---|---|---|---|---|---|---|---|---|---|
| *WHH* | 3 | 0 | 1 | 0 | 6 | 9 | 35 | 37 | 89 | 38 | 11 | 10 | 2 | 27 |
| **Persons** | | | | | | | | | | | | | | |
| *Total* | 257 | 241 | 340 | 101 | 2246 | 424 | 1591 | 1648 | 4244 | 1944 | 703 | 752 | 879 | 2360 |
| *Male* | 130 | 126 | 188 | 51 | 1142 | 227 | 786 | 810 | 2087 | 1010 | 351 | 373 | 434 | 1218 |
| *Female* | 127 | 115 | 152 | 50 | 1104 | 197 | 805 | 838 | 2157 | 934 | 352 | 379 | 445 | 1142 |
| *Sex Ratio* | 977 | 913 | 809 | 980 | 967 | 868 | 1024 | 1035 | 1034 | 925 | 1003 | 1016 | 1025 | 938 |
| **Family Size** | | | | | | | | | | | | | | |
| *Total* | 5.14 | 4.08 | 4.30 | 4.81 | 4.78 | 3.85 | 4.24 | 3.85 | 4.51 | 4.44 | 4.56 | 4.35 | 4.60 | 4.90 |
| *Hindu* | 4.69 | 4.10 | 4.30 | 4.85 | 4.79 | 4.00 | 0.62 | 0.95 | 4.52 | 4.36 | 4.29 | 4.32 | 4.60 | 4.88 |
| *Muslim* | 5.65 | 3.00 | 0.00 | 4.00 | 5.00 | 3.64 | 4.17 | 3.33 | 3.95 | 4.92 | 4.87 | 5.50 | 5.00 | 5.04 |
| *Christain* | 0.00 | 0.00 | 0.00 | 0.00 | 0.00 | 3.33 | 4.50 | 0.00 | 0.00 | 0.00 | 4.25 | 0.00 | 0.00 | 0.00 |
| *Sikh* | 5.00 | 0.00 | 0.00 | 0.00 | 3.83 | 0.00 | 2.67 | 3.33 | 0.00 | 2.00 | 5.33 | 0.00 | 0.00 | 0.00 |

| Slums | Narwal | Friends Colony | Trikuta Nagar | Dilli Channi | Noor Abad Sainik Colony | Sainik Colony Mill | Gangyal | Paloora | Uday Walla | Muthi Bridge | Greater Kalash | Balli Charan | Rajive Colony Ambadkar Mohalla | Khan Pore Sheri Kasmir Colony |
|---|---|---|---|---|---|---|---|---|---|---|---|---|---|---|
| **Households** | 592 | 89 | 54 | 338 | 125 | 86 | 116 | 61 | 50 | 138 | 183 | 255 | 777 | 508 |
| *% Hindu* | 73.48 | 100.00 | 94.44 | 94.38 | 90.40 | 100.00 | 99.14 | 68.85 | 86.00 | 100.00 | 98.36 | 80.39 | 81.34 | 3.54 |
| *% Muslim* | 26.52 | 0.00 | 5.56 | 5.62 | 8.00 | 0.00 | 0.86 | 31.15 | 14.00 | 0.00 | 1.64 | 18.04 | 15.44 | 96.26 |
| *% Christian* | 0.00 | 0.00 | 0.00 | 0.00 | 0.80 | 0.00 | 0.00 | 0.00 | 0.00 | 0.00 | 0.00 | 0.00 | 1.42 | 0.20 |
| *% Sikh* | 0.00 | 0.00 | 0.00 | 0.00 | 0.80 | 0.00 | 0.00 | 0.00 | 0.00 | 0.00 | 0.00 | 1.57 | 1.67 | 0.00 |
| *WHH* | 18 | 2 | 7 | 7 | 2 | 2 | 5 | 1 | 1 | 2 | 2 | 8 | 25 | 39 |
| **Persons** | | | | | | | | | | | | | | |
| *Total* | 2759 | 442 | 228 | 1482 | 586 | 339 | 362 | 285 | 219 | 606 | 835 | 1162 | 3393 | 2332 |
| *Male* | 1412 | 218 | 111 | 768 | 304 | 174 | 258 | 149 | 122 | 315 | 423 | 599 | 1759 | 1162 |
| *Female* | 1347 | 224 | 117 | 714 | 282 | 165 | 251 | 136 | 97 | 291 | 412 | 563 | 1634 | 1170 |
| *Sex Ratio* | 954 | 1028 | 1054 | 930 | 928 | 948 | 973 | 913 | 795 | 924 | 974 | 940 | 929 | 1007 |

Table A2. *Cont.*

| Slums | Panjtirthi | Lower Gujar Nagar | Gumat Rang Nath Mandir | Old Qasim Nagar | Bawi Basti Dograhel | Kabir Basti B C Road | Mazdoor Basti | Bagwati nagar | Bikram Chowk | Nawabad | Panama Chowk | Indra Colony Shastri Nagar | Bhawani Nagar | Gole Panjpeer |
|---|---|---|---|---|---|---|---|---|---|---|---|---|---|---|
| | | | | | | **Family Size** | | | | | | | | |
| *Total* | 4.66 | 4.97 | 4.22 | 4.38 | 4.69 | 3.94 | 3.12 | 4.67 | 4.38 | 4.39 | 4.56 | 4.56 | 4.37 | 4.59 |
| *Hindu* | 4.49 | 4.97 | 4.25 | 4.37 | 4.68 | 3.94 | 4.36 | 4.76 | 4.23 | 4.39 | 4.52 | 4.32 | 4.38 | 4.50 |
| *Muslim* | 5.14 | 0.00 | 3.67 | 4.63 | 4.70 | 0.00 | 8.00 | 4.47 | 5.29 | 0.00 | 7.00 | 5.50 | 4.33 | 4.60 |
| *Christian* | 0.00 | 0.00 | 0.00 | 0.00 | 6.00 | 0.00 | 0.00 | 0.00 | 0.00 | 0.00 | 0.00 | 0.00 | 4.27 | 5.00 |
| *Sikh* | 0.00 | 0.00 | 0.00 | 0.00 | 4.00 | 0.00 | 0.00 | 0.00 | 0.00 | 0.00 | 0.00 | 6.00 | 4.62 | 0.00 |

| Slums | Raika | Dhoke Waziran | Ganesh Bihar Muthi | Raipur Satwari | Balicharan Bagwati Nagar | Mahasha Colony | Gol Gujral Gujar Basti | Hakal | Kiryani Tallab | Tool Post Nagrota | Govind Pur Airport | Chatta Mill |
|---|---|---|---|---|---|---|---|---|---|---|---|---|
| **Households** | 34 | 84 | 143 | 287 | 136 | 152 | 23 | 69 | 536 | 139 | 95 | 78 |
| *% Hindu* | 0.00 | 94.05 | 87.41 | 93.73 | 0.00 | 100.00 | 4.35 | 100.00 | 2.43 | 100.00 | 26.32 | 89.74 |
| *% Muslim* | 100.00 | 0.00 | 12.59 | 0.35 | 100.00 | 0.00 | 95.65 | 0.00 | 97.01 | 0.00 | 1.05 | 10.26 |
| *% Christian* | 0.00 | 5.95 | 0.00 | 0.00 | 0.00 | 0.00 | 0.00 | 0.00 | 0.19 | 0.00 | 0.00 | 0.00 |
| *% Sikh* | 0.00 | 0.00 | 0.00 | 5.92 | 0.00 | 0.00 | 0.00 | 0.00 | 0.37 | 0.00 | 72.63 | 0.00 |
| **WHH** | 3 | 5 | 12 | 35 | 2 | 7 | 1 | 6 | 49 | 5 | 5 | 4 |
| | | | | | | **Persons** | | | | | | |
| *Total* | 166 | 388 | 621 | 1159 | 634 | 545 | 121 | 292 | 2248 | 653 | 398 | 323 |
| *Male* | 82 | 198 | 338 | 591 | 338 | 273 | 64 | 155 | 1163 | 330 | 206 | 169 |
| *Female* | 84 | 190 | 283 | 568 | 296 | 272 | 57 | 137 | 1085 | 323 | 192 | 154 |
| *Sex Ratio* | 1024 | 960 | 837 | 961 | 876 | 996 | 891 | 884 | 933 | 979 | 932 | 911 |
| | | | | | | **Family Size** | | | | | | |
| *Total* | 4.88 | 4.62 | 4.34 | 4.04 | 4.66 | 3.59 | 5.26 | 4.23 | 4.19 | 4.70 | 4.19 | 4.14 |
| *Hindu* | 0.00 | 4.67 | 4.18 | 4.07 | 0.00 | 0.00 | 6.00 | 4.23 | 4.31 | 4.70 | 4.00 | 4.11 |
| *Muslim* | 4.88 | 0.00 | 5.44 | 5.00 | 4.66 | 0.00 | 5.23 | 0.00 | 4.20 | 0.00 | 4.00 | 4.38 |
| *Christian* | 0.00 | 3.80 | 0.00 | 0.00 | 0.00 | 0.00 | 0.00 | 0.00 | 3.00 | 0.00 | 0.00 | 0.00 |
| *Sikh* | 0.00 | 0.00 | 0.00 | 3.53 | 0.00 | 0.00 | 0.00 | 0.00 | 3.00 | 0.00 | 4.26 | 0.00 |

**Table A3.** Percentage of households under matrix parameters for 1x1x2.

| Slum Name | Housing | | Tenure | | | Infrastructure | | |
|---|---|---|---|---|---|---|---|---|
| | % of Kutcha HH | % of Semi-Pucca HH | Secure Tenure | Insecure Tenure | %HH with In-house Water Connection | % of Pucca Roads | % of HH with Electricity | % of HH with Toilet |
| Bawi Basti Dograhel | 22.73 | 56.82 | 93.18 | 6.82 | 45.40 | 0.00 | 100.00 | 0.00 |
| Kabir Basti B C Road | 100.00 | 0.00 | 88.70 | 11.30 | 0.00 | 40.00 | 100.00 | 0.00 |
| Mazdoor Basti | 98.33 | 0.00 | 90.00 | 10.00 | 98.33 | 3.33 | 100.00 | 0.00 |
| Indra Colony Shastri Nagar | 87.60 | 12.40 | 66.67 | 33.33 | 0.78 | 0.78 | 100.00 | 0.00 |
| Bhawani Nagar | 14.29 | 80.95 | 92.86 | 7.14 | 33.33 | 23.81 | 100.00 | 0.00 |
| Gole Panjpeer | 60.08 | 35.66 | 82.95 | 17.05 | 33.33 | 29.84 | 100.00 | 0.00 |
| Omnagar | 94.00 | 6.00 | 100.00 | 0.00 | 14.00 | 54.00 | 100.00 | 2.00 |
| Pounch House | 100.00 | 0.00 | 0.00 | 100.00 | 1.69 | 16.95 | 100.00 | 0.00 |
| Mahash Basti | 74.40 | 24.80 | 89.33 | 10.67 | 45.07 | 7.47 | 99.73 | 0.27 |
| Press More Bahu Fort | 49.07 | 50.70 | 91.12 | 8.88 | 0.00 | 3.50 | 100.00 | 0.00 |
| Khan Pore Sheri Kasmir Colony | 55.51 | 29.53 | 97.64 | 2.36 | 86.61 | 0.59 | 100.00 | 2.76 |
| Raika | 88.24 | 11.76 | 79.41 | 20.59 | 0.00 | 0.00 | 100.00 | 0.00 |
| Dhoke Waziran | 45.24 | 52.38 | 98.81 | 1.19 | 4.76 | 36.90 | 100.00 | 0.00 |
| Ganesh Bihar Muthi | 72.73 | 25.17 | 97.20 | 2.80 | 74.83 | 4.90 | 100.00 | 0.70 |
| Raipur Satwari | 42.86 | 46.34 | 94.08 | 5.92 | 41.81 | 12.20 | 100.00 | 0.00 |
| Mahasha Colony | 84.21 | 15.79 | 82.24 | 17.76 | 0.00 | 0.66 | 100.00 | 0.00 |
| Gol Gujral Gujar Basti | 100.00 | 0.00 | 100.00 | 0.00 | 0.00 | 34.78 | 100.00 | 0.00 |
| Hakal | 86.96 | 13.04 | 100.00 | 0.00 | 88.41 | 4.35 | 100.00 | 0.00 |
| Kiryani Tallab | 98.32 | 1.31 | 3.73 | 96.27 | 0.37 | 13.06 | 99.63 | 0.19 |
| Govind Pur Airport | 86.32 | 13.68 | 87.37 | 12.63 | 43.16 | 10.53 | 98.95 | 1.05 |
| Chatta Mill | 62.82 | 37.18 | 83.33 | 16.67 | 98.72 | 0.00 | 100.00 | 2.56 |

Table A4. Percentage of households under matrix parameters for 1 × 2 × 2.

| Slum Name | Housing | | Tenure | | | Infrastructure | | |
|---|---|---|---|---|---|---|---|---|
| | % of Kutcha HH | % of Semi-Pucca HH | Secure Tenure | Insecure Tenure | % HH with in-house Water Connection | % of Pucca Roads | % of HH with Electricity | % of HH with Toilet |
| Panjtirthi | 100.00 | 0.00 | 0.88 | 99.12 | 0.00 | 10.62 | 100.00 | 0.00 |
| Lower Gujar Nagar | 93.75 | 6.25 | 0.00 | 100.00 | 0.00 | 6.25 | 100.00 | 0.00 |
| Gumat Rang Nath Mandir | 98.36 | 1.64 | 0.00 | 100.00 | 0.00 | 3.28 | 100.00 | 0.00 |
| Old Qasim Nagar | 56.41 | 43.59 | 0.00 | 100.00 | 10.26 | 46.15 | 100.00 | 2.56 |
| Bagwatinagar | 99.44 | 0.56 | 0.85 | 99.15 | 0.00 | 40.68 | 100.00 | 0.00 |
| Bikram Chowk | 66.34 | 33.66 | 10.40 | 89.60 | 50.50 | 4.46 | 99.50 | 0.00 |
| Nawabad | 100.00 | 0.00 | 0.00 | 100.00 | 2.33 | 0.00 | 100.00 | 0.00 |
| Panama Chowk | 93.59 | 6.41 | 21.79 | 78.21 | 2.56 | 65.38 | 100.00 | 2.56 |
| Suraj Nagar Tallu Tallab | 98.73 | 1.27 | 0.00 | 100.00 | 2.53 | 36.71 | 100.00 | 1.27 |
| Zorawar Singh Chowk | 100.00 | 0.00 | 0.00 | 100.00 | 0.00 | 0.00 | 100.00 | 0.00 |
| Preet Nagar Digyani | 99.57 | 0.43 | 0.43 | 99.57 | 0.64 | 39.57 | 100.00 | 0.00 |
| Bahu Fort | 68.18 | 31.82 | 58.18 | 41.82 | 0.00 | 9.09 | 100.00 | 0.00 |
| Rajiv Nagar | 66.99 | 32.80 | 2.65 | 97.35 | 0.21 | 0.96 | 99.68 | 0.11 |
| Kalka Colony Qassim Nagar | 97.72 | 2.05 | 2.28 | 97.72 | 0.46 | 28.54 | 99.77 | 0.23 |
| Peer Baba Rakh Bahu | 77.92 | 22.08 | 1.95 | 98.05 | 3.90 | 35.06 | 100.00 | 1.30 |
| Bhalla Colony | 91.33 | 8.67 | 0.00 | 100.00 | 0.00 | 54.34 | 100.00 | 0.00 |
| Channi Rama | 100.00 | 0.00 | 0.00 | 100.00 | 0.52 | 48.17 | 98.95 | 0.00 |
| Marathi Mohalla Railway | 99.17 | 0.83 | 0.83 | 99.17 | 0.21 | 8.09 | 99.59 | 0.00 |
| Narwal | 99.49 | 0.51 | 0.68 | 99.32 | 2.20 | 37.84 | 100.00 | 0.17 |
| Friends Colony | 100.00 | 0.00 | 0.00 | 100.00 | 12.36 | 71.91 | 100.00 | 1.12 |
| Trikuta Nagar | 98.15 | 1.85 | 0.00 | 100.00 | 0.00 | 98.15 | 100.00 | 1.85 |
| Dilli Channi | 89.35 | 10.65 | 0.59 | 99.41 | 0.59 | 90.83 | 100.00 | 0.00 |
| Noor Abad Sainik Colony | 89.60 | 10.40 | 1.60 | 98.40 | 0.80 | 17.60 | 100.00 | 0.00 |
| Sainik Colony Mill | 98.84 | 1.16 | 0.00 | 100.00 | 3.49 | 53.49 | 100.00 | 6.98 |
| Gangyal | 100.00 | 0.00 | 0.00 | 100.00 | 1.72 | 1.72 | 100.00 | 0.00 |
| Paloora | 100.00 | 0.00 | 0.00 | 100.00 | 0.00 | 34.43 | 100.00 | 0.00 |

**Table A4.** *Cont.*

| Slum Name | Housing | | Tenure | | | Infrastructure | | |
|---|---|---|---|---|---|---|---|---|
| | % of Kutcha HH | % of Semi-Pucca HH | Secure Tenure | Insecure Tenure | % HH with in-house Water Connection | % of Pucca Roads | % of HH with Electricity | % of HH with Toilet |
| Uday Walla | 100.00 | 0.00 | 0.00 | 100.00 | 2.00 | 76.00 | 98.00 | 2.00 |
| Muthi Bridge | 97.10 | 2.90 | 5.07 | 94.93 | 0.00 | 15.22 | 100.00 | 0.00 |
| Greater Kalash | 90.71 | 3.83 | 0.00 | 100.00 | 0.00 | 68.85 | 100.00 | 0.00 |
| Balli Charan | 95.69 | 4.31 | 4.31 | 95.69 | 0.00 | 0.39 | 100.00 | 0.00 |
| Rajive Colony Ambadkar Mohalla | 80.05 | 19.95 | 9.91 | 90.09 | 0.90 | 16.47 | 100.00 | 0.13 |
| Balicharan Bagwati Nagar | 94.12 | 2.94 | 0.74 | 99.26 | 56.62 | 0.00 | 63.24 | 0.00 |
| Tool Post Nagrota | 100.00 | 0.00 | 0.72 | 99.28 | 2.88 | 18.71 | 98.56 | 0.72 |

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
