# Peer review of "Strategic Slum Upgrading and Redevelopment Action Plan for Jammu City"

_resources, doi:10.3390/resources11120120_

Round 1

Reviewer 1 Report

This is an interesting paper about slum upgrading methods that are necessary in developing countries. The authors took advantage of the analysis from MIS and GIS technologies which help to prioritize each of the slum areas. The results can be a good implication for the public policy of slum upgrading. However, this paper needs to revise something critical to be published as an academic paper. Please refer to my detailed comments below.

1.         This paper does not have any section of the literature review. Thus, I cannot find any research gap and literature contribution. This is a crucial weakness of this research to be an academic paper. Authors noted “there is no universally accepted strategy for improving their conditions. (Line 23-24)“ but I do not agree with the statement. The authors should review all strategies to find the novelty of this study. Many related slum studies including international slum upgrading programs can be found in the literature since the slum study has a long history. Thus, the authors should extensively update this paper with the literature review.

2.         Section 1(Introduction) is well organized with proper information. But I am not clear about the statement (lines 70-72). This seems like a slum definition but no citation for the factor like “a minimum of 300 people and roughly 60-70 households ….” How did the authors define the attributes? Please clarify.

3.         Section 2 looks research methods. The authors note general information about Jammu at the beginning (lines 114-131) but they need to highlight why the authors selected Jammu for this study. Why not other slum cities in India for this study? The authors need to clarify.

4.         Figure 1 and 3 need any source information,

5.         What is LULC (line 140)? A full name is needed.

6.         Authors note “geo-physical and socio-economic considerations were central….”. Why only 2 categories? Please clarify the reasons.

7.         (line152-153) “The dataset collected from the field survey(tenure status, tenability, ownership, dwelling density)” Authors need to describe the procedure detail of the field survey for the data validity and reliability. Period? Any sampling strategy? Is any resident interview based on approval? etc...

8.         (Line174) authors selected three categories (Infrastructure, housing, and tenure status) to determine which areas are most in need of rapid improvement. Why not ‘dwelling density’(your figure 2 also shows ‘dwelling density’)? The density is one of the most critical factors to determine the slum environment as well (UN-habitat also indicates)

9.         (Line 179) what are “kuccha” houses and “semi-pucca”? they seem local terms. Please clarify them for international readers. Table 1 shows “Kaccha”. Which is the correct spelling?

10.       (Line 183) Why 2*2*2 matrix? Isn’t it too simplified? Why not 3 matrix system for more accuracy and specification?

11.       (Line 184-185) The authors note that the rating system was based on studies and community conversations. What studies? Need proper citations. What community conversations? Please clarify.

12.      (Table 1) the deficiency level of the Infrastructure part is divided into 60. But tenure and housing parts are divided into 40. Why different? Why not 50 for all 3 parts? Please clarify.

13.      Section 3 (results and discussions) needs connections with the literature review for rich academic discussions. Then the novelty and originality can be highlighted in this paper. 

14.      Section 4 (conclusions) needs study limitations.

Author Response

#Review Report 1 Comments & Suggestions for Authors

Comment: This is an interesting paper about slum upgrading methods that are necessary in developing countries. The authors took advantage of the analysis from MIS and GIS technologies which help to prioritize each of the slum areas. The results can be a good implication for the public policy of slum upgrading. However, this paper needs to revise something critical to be published as an academic paper. Please refer to my detailed comments below.

Response: First of all, we are grateful to the respected reviewer for encouraging and positive comments on our manuscript. We really appreciate the suggestions given by the reviewer for the improvement of our manuscript. In line with the suggestions, we have incorporated the changes in the revised manuscript. The response to the specific comments is given below

Comment 1: This paper does not have any section of the literature review. Thus, I cannot find any research gap and literature contributions. This is a crucial weakness of this research to be an academic paper. Authors noted “there is no universally accepted strategy for improving their conditions. (Line 23-24)“ but I do not agree with the statement. The authors should review all strategies to find the novelty of this study. Many related slum studies including international slum upgrading programs can be found in the literature since the slum study has a long history. Thus, the authors should extensively update this paper with the literature review.

Response: Literature review incorporated in revised manuscript. The statement at line 23-24 modified as per the suggestion of a respected reviewer.

Comment 2: Section 1(Introduction) is well organized with proper information. But I am not clear about the statement (lines 70-72). This seems like a slum definition but no citation for the factor like “a minimum of 300 people and roughly 60-70 households ….” How did the authors define the attributes? Please clarify

Response: In revised manuscript we have included the citation for the statement and these figures have been quoted as per the “Ministry of Housing and Urban Poverty Alleviation GoI” census report. The census of India has given different definitions for Slums as per usage of each agency and individual.

Comment 3: Section 2 looks at research methods. The authors note general information about Jammu at the beginning (lines 114-131) but they need to highlight why the authors selected Jammu for this study. Why not other slum cities in India for this study? The authors need to clarify.

Response: In J&K only, Jammu is having maximum number of slums and is going through the rapid development phase including the implementation of smart city program. Since our domain of research is within J&K only and keeping in view the requirements of the local government only Jammu City has been taken up for the study.

In revised manuscript the clarification regarding the selection of this particular city has been incorporated.

Comment 4: Figure 1 and 3 need any source information

Response: Both the figures have been prepared by the authors themselves, not obtained from any source. The details about the procurement of ward boundaries have been mentioned in the revised manuscript. i.e “Jammu Municipal Corporation”.

Comment 5: What is LULC (line 140)? A full name is needed.

Response: Full name of LULC has been mentioned in the revised manuscript.

Comment 6: Authors note “geo-physical and socio-economic considerations were central….”. Why only 2 categories? Please clarify the reasons.

Response: The majority of the research on slums going on, mainly focuses on one of three constructs: exploring the socio-economic and policy issues(e.g., Omole, 2010; Patel, Koizumi, & Crooks, 2014; Sola, 2013); exploring the physical characteristics (e.g., Filho & Sobreira, 2005; Kit, Lüdeke, & Reckien, 2012; Kohli, Sliuzas, Kerle, & Stein, 2012); and, lastly, those modeling slums (e.g., Augustijn-Beckers, Flacke, & Retsios, 2011; Jokar Arsanjani, Helbich, Kainz, & Darvishi Boloorani, 2013; Patel, Crooks, & Koizumi, 2012; Sietchiping, 2004).

This paper incorporated one among the above-mentioned approaches which was valuable as per the requirement of the local government and funding patterns. By using the social and physical constructs, this paper provides a more holistic synthesis of the problem for immediate intervention, which can potentially lead to a deeper understanding and, consequently, better approaches for tackling the challenge of slums at local scales.

In the light of the framework put forward, the main contribution of this paper is twofold: first, it identifies the most pressing issues surrounding slums based on available time frame and funding; and second, it puts forward a future integrated research agenda for developing a deeper understanding of the fundamental underlying processes that define and shape slums.

Comment 7: (line152-153) “The dataset collected from the field survey (tenure status, tenability, ownership, dwelling density)” Authors need to describe the procedure detail of the field survey for the data validity and reliability. Period? Any sampling strategy? Is any resident interview based on approval? Etc

Response: Comprehensive list of all slums (notified, non-notified, recognized and identified) on lands belonging to State / Central Government, urban local bodies, public undertakings of State / Central Government, any other public agency and private land should be collected from the concerned departments. After finalization of the slum list, a unique slum code (e.g. 001, 002….) for each slum has been generated as per the guidelines of National Buildings Organisation (NBO) which is under the Ministry of Housing and Urban Affair. The boundary of each slum will be marked on the GIS base map of the planning area with the help of satellite image and using GPS. The GIS and MIS data has been integrated using the unique slum code for linking databases on socio-economic, tenability status, land tenure, land ownership and land value of the slums.  The interview of slum dwellers was conducted with the approval of the local government. No sampling-based survey was conducted. Each and every household was surveyed during the field survey.

Comment 8: (Line174) authors selected three categories (Infrastructure, housing, and tenure status) to determine which areas are most in need of rapid improvement. Why not ‘dwelling density’(your figure 2 also shows ‘dwelling density’)? The density is one of the most critical factors to determine the slum environment as well (UN-habitat also indicates).

Response: As per the guidelines formulated by Rajiv Awas Yogna (RAY) we have used three main categories in this study.  For further detailed analysis and extraction of more details like tenable slums, dwelling density was calculated from a developmental point of view.  It is important to represent the dwelling density in form of spatial maps so that planners can get an idea which area is having high, medium or low density for planning purposes.

No of Dwelling Units (DUs) /Hectare (ha) was calculated based on the data obtained from the Slum Survey Profile, as per NBO format. The following density norms for net residential density are suggested based on IS 8888 standards.

High >500 DU/ha4

Medium 350-500 DU/ha

Low < 350 DU/ha

Comment 9: (Line 179) what are “kuccha” houses and “semipucca”? they seem local terms. Please clarify them for international readers. Table 1 shows “Kaccha”. Which is the correct spelling?

Response: The descriptions of both the terms have been mentioned in the revised manuscript at first place in text. The spelling of “Kuccha” has been replaced with correct one “Kutcha”.

Comment 10: (Line 183) Why 2*2*2 matrix? Isn’t it too simplified? Why not 3 matrix systems for more accuracy and specification?

Response: The priority matrix has been used in this study for ranking the severity and prioritizing the slums into two classes i.e High and Low priority for intervention by local government. The rationale behind selecting the priority matrix (2x2) in this study is due to the fact that this method helps to reduce the data to simple variables for categorizing slums for qualitative judgements based on availability of data. The method helped in the initial sorting of qualitative data which further helped to define the priorities of the slums for proper planning and management.

The matrix was found to be suitable because the overall number of the slums in Jammu is very low as compared to the major metropolitan cities in India. Based on the availability of funds the local government had to prioritize only a few slums on the basis of deficiencies, therefore keeping in view the less number of slums and corresponding budget availability 2x2 matrix was used to categorize the slums as High and Low Priority. The medium priority option was not taken into consideration due to the intended aim of local government for prioritization of slums.

Comment 11: (Line 184-185) The authors note that the rating system was based on studies and community conversations. What studies? Need proper citations. What community conversations? Please clarify.

Response: The citations have been included in revised manuscript against the statement.

During the field survey of slum areas, the questionnaire was designed in a way so that information will be collected from slum dwellers about their living conditions. Based on such gathered information (community response) the rating system was decided keeping in view which factor/indicator contributes more towards the deteriorating condition of slum dwellers. The indicators were ranked in a way so that it will be easy to decide for the governmental agencies which slum area needs immediate intervention.

Comment 12: (Table 1) the deficiency level of the Infrastructure part is divided into 60. But tenure and housing parts are divided into 40. Why different? Why not 50 for all 3 parts? Please clarify.

Response: The main aim behind this study was the upgradation of existing slum dwellers and formulation of redevelopment plan. The budget has been allocated by the Government in a way so that the living standards of existing slum dwellers will be given high priority. The interview/conservations with slum communities were taken into consideration while deciding the indicators and their importance in prioritization context. Based on the field survey it was observed that almost all the slum dwellers have tenure and housing facilities but their living conditions can be improved if the basic infrastructural facilities/amenities will be provided to them. This is the reason for allocating the high percentage in a scale of 0-100 to infrastructure and equal percentage to tenure and housing. This technique helped the government agencies to work toward the betterment of existing slum dwellers. If it would have been the equal weightage system that will not help to identify the most important parameter which needs immediate intervention.

Comment 13: Section 3 (results and discussions) needs connections with the literature review for rich academic discussions. Then the novelty and originality can be highlighted in this paper.

Response: Literature have been cited in results and discussion section in revised manuscript.

Comment 14: Section 4 (conclusions) needs study limitations.

Response: Although this study provides promising initial evidence on the informal settlements and their conditions in Jammu city, there is a lack of some important aspects which can be taken into consideration in future to overcome the limitations of this study.  Rather than simply measuring the basic infrastructural and amenities, the analysis can be done at broader scale incorporating ladders that were recently proposed by the WHO/UNICEF Joint Monitoring Programme to further differentiate between households on the basis of levels of service quality.

In its present form the study has not focused on vulnerability of slums with respect to natural disasters. In future, we intend to take up the household level risk and vulnerability assessments using descriptive statistics to analyze the status and drivers for mainstreaming the adaptation and mitigation strategies.

Reviewer 2 Report

Abstract: perhaps use another word for 'unceremonious'.

'The deliverable is'-it the aim of this research?

Line 62: Explanation here of what slums are should be earlier in the paper

line 140: Write full form for LULC the first time it is used

line 179: explain what kuccha and pucca means

Some sections can be shortened such as: details about Jammu. If that information has no bearing on the results or used in discussion, it can be removed.

Not very clear why this method was chosen; how it compares to other methods or how it works

There is a lot of information but not much analysis of the data. there has to be high level interpretation of the data and then an example.

The argument for the article's contribution and why this research fills a gap is not strong.

Author Response

#Review Report 2 Comments & Suggestions for Authors

Comment 1: Abstract: perhaps use another word for 'unceremonious'.

Response: The word has been replaced by “Squatter” in the revised manuscript.

Comment 2: 'The deliverable is'-it the aim of this research?

Response: Yes, it is the aim of the study and modified the text in revised manuscript.

Comment 3: Line 62: Explanation here of what slums are should be earlier in the paper

Response: As per the suggestion of the reviewer the explanation of slums have been mentioned earlier in paper.

Comment 4: line 140: Write full form for LULC the first time it is used

Response: Full form of LULC mentioned.

Comment 5: line 179: explain what kuccha and pucca means

Response: There is a mistake in spelling. Rectified in revised manuscript. The descriptions of both the terms mentioned at first place in text.

Comment 6: Some sections can be shortened such as: details about Jammu. If that information has no bearing on the results or used in discussion, it can be removed.

Response: The details mentioned about Jammu cannot be shortened because the information has bearing on the results.

Comment 7: Not very clear why this method was chosen; how it compares to other methods or how it works

Response: The priority matrix has been used in this study for ranking the severity and prioritizing the slums into two classes i.e High and Low priority for intervention by local government. The rationale behind selecting the priority matrix (2x2) in this study is due to the fact that this method helps to reduce the data to simple variables for categorizing slums for qualitative judgements based on availability of data. The method helped in the initial sorting of qualitative data which further helped to define the priorities of the slums for proper planning and management.

The matrix was found to be suitable because the overall number of the slums in Jammu is very low as compared to the major metropolitan cities in India. Based on the availability of funds the local government had to prioritize only a few slums on the basis of deficiencies, therefore keeping in view the smaller number of slums and corresponding budget availability 2x2 matrix was used to categorize the slums as High and Low Priority. The medium priority option was not taken into consideration due to the intended aim of local government for prioritization of slums.

Comment 8: There is a lot of information but not much analysis of the data there has to be high level interpretation of the data and then an example.

Response: Thank you for this valuable suggestion. We will definitely try to incorporate more analysis in our future study. We intend to use descriptive statistics like proportions, means, standard deviations etc for indicator characteristics focusing on essential utilities on a micro scale level (household level) for future assessments. As of now given the requirement of the project the analysis was limited to generation of prioritization matrix only.

Comment 9: The argument for the article's contribution and why this research fills a gap is not strong.

Response: The introduction section has been improved in line of the suggestions given by respected reviewer

Round 2

Reviewer 1 Report

The paper seems more improved with additional descriptions and clarification. There are just minor comments to publish. 

1. During reading the response letter, It was hard to find where the author's defense contents are located in the revised manuscript due to missing line numbers in the response letter. Hopefully, there are indications to make clear connections. 

2. Figures should be updated for better readability 

- Figure 4: too small texts 

- Figure 6, 10, 12: the line colors are not clear so hard to recognize each factor on the map. The lines can be thickened with clear colors. 

- Figure 8: I can not read the texts on the image 

3. The conclusion is generally ok but authors should clearly highlight the novelty and originality of the study findings in the context of the literature. In addition, the policy implications can be also more underlined. 

Author Response

#Review Report 1 Comments & Suggestions for Authors

Comment: The paper seems more improved with additional descriptions and clarification. There are just minor comments to publish. 

Response: Thank you for your positive response. Your comments have really helped in refining the manuscript. We have further improved the manuscript in light of minor comments.

Comment 1: During reading the response letter, It was hard to find where the author's defense contents are located in the revised manuscript due to missing line numbers in the response letter. Hopefully, there are indications to make clear connections. 

Response: The corrections made in the revised manuscript are highlighted in yellow color. Since the order of line number changed significantly due to incorporations of revisions therefore mentioning the line number seemed irrelevant and would have led to confusion.

Comment 2: Figures should be updated for better readability.

Response:  Noted and all figures updated accordingly.

Comment 3: Figure 4: too small texts.

Response:  Noted and Revised.

Comment 4: Figure 6, 10, 12: the line colors are not clear so it is hard to recognize each factor on the map. The lines can be thickened with clear colors. 

Response: Noted and all figures updated accordingly.

Comment 5: Figure 8: I cannot read the texts on the image 

Response: Noted and revised to the best possible resolution, however since MIS is a screen capture we are not able to improve its resolution further.

Comment 6: The conclusion is generally ok but authors should clearly highlight the novelty and originality of the study findings in the context of the literature. In addition, the policy implications can be also more underlined. 

Response: Revised.

Reviewer 2 Report

The changes made by the authors has improved the quality of the paper. Well done

Is this sentence factually correct? if not can you please rephrase?

Line 23

"The majority of people in developing-world live in squatter settlements"....

Author Response

#Review Report 2 Comments & Suggestions for Authors

Comment: The changes made by the authors has improved the quality of the paper. Well done

Is this sentence factually correct? if not can you please rephrase?

Line 23 "The majority of people in developing-world live in squatter settlements"....

Response: Thank you to the respected reviewer for encouraging words. We cross checked the statement and it's correct, since it's talking about only developing nations, not the world as a whole.  Among 9 billion of the world’s population, 1 billion of people reside in squatter settlements (Mohanty, 2020).

(Mohanty, M. (2020). Squatter Settlements and Slums and Sustainable Development. In: Leal Filho, W., Azul, A., Brandli, L., Özuyar, P., Wall, T. (eds) Sustainable Cities and Communities. Encyclopedia of the UN Sustainable Development Goals. Springer, Cham.